# *Do* Finetti: on Causal Effects for Exchangeable Data

**Siyuan Guo**[14] *    **Chi Zhang**[2]    **Karthika Mohan**[3]    **Ferenc Huszár**[4†]    **Bernhard Schölkopf**[1†]

[1]Max Planck Institute for Intelligent Systems    [2]Toyota Research Institute
[3] Oregon State University    [4] University of Cambridge
† Equal supervision

## Abstract

We study causal effect estimation in a setting where the data are not i.i.d. (independent and identically distributed). We focus on exchangeable data satisfying an assumption of independent causal mechanisms. Traditional causal effect estimation frameworks, e.g., relying on structural causal models and do-calculus, are typically limited to i.i.d. data and do not extend to more general exchangeable generative processes, which naturally arise in multi-environment data. To address this gap, we develop a generalized framework for exchangeable data and introduce a truncated factorization formula that facilitates both the identification and estimation of causal effects in our setting. To illustrate potential applications, we introduce a causal Pólya urn model and demonstrate how intervention propagates effects in exchangeable data settings. Finally, we develop an algorithm that performs simultaneous causal discovery and effect estimation given multi-environment data.

## 1   Introduction

Inferring causal effects from observational data is a central task for scientific applications from health and epidemiology to the social and behavioral sciences. Scientists aim to understand Nature's mechanisms to discover feasible intervention targets and estimate intervention effects. The existing causal effect estimation theory is often formulated for structural causal models [Pearl, 2009], focusing on the study of *independent and identically distributed* (i.i.d.) data. Indeed, the do-calculus [Pearl, 2012] and its extensions and applications (e.g., mediation analysis [Pearl, 2022], stochastic interventions [Correa and Bareinboim, 2020]) developed in the i.i.d. framework has been widely used in practice.

A more recent line of work relaxes the i.i.d. assumption and considers causal inference under the assumption of *independent causal mechanisms* (ICM) [Schölkopf et al., 2012, Pearl, 2009, Guo et al., 2023a], which postulates that distinct causal mechanisms of the true underlying generating process do not inform or influence one another. The *Causal de Finetti* theorems of Guo et al. [2023a] provide a mathematical justification of the ICM assumption in exchangeable data-generation processes, thus establishing a foundation for studying causality in exchangeable data. Exchangeable data generation processes that adhere to the *ICM* principle are referred to as *ICM generative processes*, or *ICM-exchangeable data*.

Guo et al. [2023a] showed that data from ICM generative processes, or more succinctly ICM-exchangeable data, are more informative than i.i.d. data in that they often permit unique causal structure identification in cases where i.i.d. data does not. In contrast, the present work focuses on causal effect identification and estimation. Standard formalisms in the i.i.d. framework, e.g., using structural causal models and do-operators, do not apply in the exchangeable non-i.i.d. setting. We study the meaning of interventions, feasible intervention targets, and finally causal effect identification and estimation for ICM generative processes. In doing so, we make three contributions:

---

*Correspondence to `siyuan.guo@tuebingen.mpg.de`

38th Conference on Neural Information Processing Systems (NeurIPS 2024).

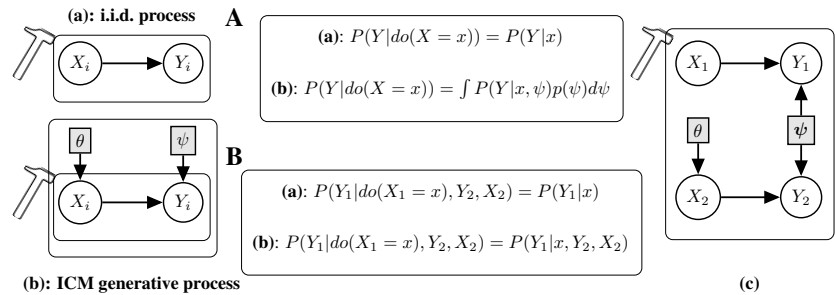

Figure 1: A bivariate illustration demonstrates differences in causal effect between i.i.d. processes and ICM-generative processes. Suppose $\mathcal{G} = X \rightarrow Y$. The hammer represents an intervention on the closest node. (a): Data generated according to $\mathcal{G}$ under an i.i.d. process; (b): Data generated under an ICM-generative process (using plate notation). Block **A** shows how $P(Y|do(X))$ differs between the i.i.d. (a) and the exchangeable (b) case. Note that the causal effect under *i.i.d.* is a special case of that under *exchangeable* processes with $p(\psi) = \delta(\psi = \psi_0)$, for some value $\psi_0$. Corollary 1 below justifies that we omit position indices from Block **A** in ICM-generative processes. Block **B** shows the difference in intervention effect when conditioned on other observations, for i.i.d. (a) and ICM-generative (b) processes. In the i.i.d. case (a), due to $(Y_1, X_1) \perp\!\!\!\perp (Y_2, X_2)$, conditioning on $(X_2, Y_2)$ conveys no information on the prediction of the interventional effect on $Y_1$. In contrast, for an ICM-generative process (b), observing $(X_2, Y_2)$ does provide additional information about the effect on $Y_1$ when intervening on $X_1$. Graph (c) illustrates the graph surgery performed on ICM($\mathcal{G}$) (cf. Def. 7 in Appendix A). We observe that conditioning on a collider $Y_2$ provides additional information about the effect on $Y_1$ when intervening on $X_1$.

- **Causal effect in ICM generative processes (Section 3)**: We establish the operational meaning of interventions and the feasible intervention targets in ICM generative processes that differ from the traditional framework in i.i.d. data.

- Our main **causal effect identification and estimation theorem (Section 3.1)** provides an explicit truncated factorization formula for ICM generative processes that solves the problems of identification and estimation of causal effects. In particular, we introduce a **causal Pólya urn model (Section 3.2)** and show that the post-interventional distribution changes when conditioning on other observations – a property that does not exist in i.i.d. data.

- **Causal effect estimation in multi-environment data (Section 4)**: we connect ICM processes with multi-environment data, showing that extending the causal framework to exchangeable data does not necessarily mean a decrease of its ability in graphical identification and effect estimation. Theorem 2, informally stated, shows that with no prior graphical assumptions, data adhering to the ICM principle and generated by an exchangeable process alone is sufficient for both graphical and effect identification. We developed a *Do-Finetti* algorithm for multi-environment data, and empirically validate our results in Section 5.

Fig. 1 illustrates the main differences in causal effects between i.i.d. and ICM generative processes.

## 2 Preliminaries

**Notation** Denote $X$ as a random variable with realization $x$ with boldsymboled $\mathbf{X}$ as a set of random variables with realizations $\mathbf{x}$. Data generated by either i.i.d. or ICM generative processes is a sequence of random variables $\mathbf{X}_{:;1}, \mathbf{X}_{:;2}, \mathbf{X}_{:;3}, \ldots$, where $\mathbf{X}_{:;n} := (X_{1;n}, \ldots, X_{d;n})$ and $d$ denotes the number of variable indices and $n$ denotes the position index within a sequence. For example, $X_{i;n}$ denotes the $i$-th random variable observed at the $n$-th position in a sequence. Often in this work, $N$ denotes the number of positions observed in a sequence and $[N] := \{1, \ldots, N\}$ as the set of positive integers less than or equal to $N$ and $[\neg\mathbf{S}]$ denotes the set of positive integers less than equal to $N$ excluding values contained in $\mathbf{S}$. To abbreviate notation, write $\mathbf{X}_{i;[N]} := (X_{i;1}, \ldots, X_{i;N})$. For simplicity of illustration, we use bivariate examples with $X_n, Y_n$ as a pair of random variables observed at the $n$-th position. Putting the bivariate notation in the above multivariate context, $X_{1;n}$ corresponds to $X_n$ and $X_{2;n}$ corresponds to $Y_n$ where the first index indicates different variables in the same position. Denote uppercase $P$ for probability distribution and lowercase $p$ for probability density functions.

## 2.1 The Causal Framework in i.i.d. Data

**Structural Causal Model** [Peters et al., 2017, Pearl, 2009, Haavelmo et al., 1995, Wright, 1921] A structural causal model (SCM) $\mathcal{M} := (\mathbf{F}, P_{\mathbf{U}})$ consists of a set of structural assignments $\mathbf{F} := \{f_1, \ldots, f_d\}$ such that $X_j := f_j(\mathbf{PA}_j, U_j), j = 1, \ldots, d$, where $\mathbf{PA}_j \subseteq \{X_1, \ldots, X_d\}\backslash\{X_j\}$ and are often called as parents of $X_j$. The joint distribution $P_{\mathbf{U}}$ over the noise or exogenous variables $\mathbf{U}$ is assumed to be jointly independent. When such a condition is satisfied, the SCM is a Markovian model that assumes the absence of unmeasured confounders. A graph $\mathcal{G}$ of an SCM is obtained by creating one vertex $X_j$ and drawing directed edges from each parent in $\mathbf{PA}_j$ to $X_j$. We assume $\mathcal{G}$ is acyclic. Our lack of knowledge in noise variables $P_{\mathbf{U}}$ and together with the structural assignments $\mathbf{F}$ induces a joint distribution over the observable variables $P(X_1, \ldots, X_d)$. Such joint distribution satisfies the *Markov factorization property* with respect to $\mathcal{G}$:

$$P(X_1, \ldots, X_d) = \prod_{i=1}^{d} P(X_i \mid \mathbf{PA}_i), \tag{1}$$

Given two disjoint sets of variables $\mathbf{X}$ and $\mathbf{Y}$, the **causal effect** of $\mathbf{X}$ on $\mathbf{Y}$, denoted as $P(\mathbf{Y}|do(\mathbf{X}))$, is defined with respect to modifications of an existing SCM (also known as *graph surgery*): for each realization $\mathbf{x}$ of $\mathbf{X}$, $P(\mathbf{y}|do(\mathbf{x}))$ gives the probability of $\mathbf{Y} = \mathbf{y}$ induced by deleting from the SCM all structural assignments corresponding to variables in $\mathbf{X}$ and substituting $\mathbf{X} = \mathbf{x}$ in the remaining equations. In Markovian models, given a graph, causal effect is identifiable via Eq. (2):

$$P(X_1, \ldots, X_d|do(\mathbf{X} = \mathbf{x})) = \prod_{i:X_i \notin \mathbf{X}} P(X_i|\mathbf{PA}_i)\big|_{\mathbf{X}=\mathbf{x}}, \tag{2}$$

where $|_{\mathbf{X}=\mathbf{x}}$ enforces $X_1, \ldots, X_d$ to be consistent with realizations of $\mathbf{X}$ else Eq. (2) takes value 0. This principled approach is known as *g-computation formula* [Robins, 1986], *truncated factorization* [Pearl, 2009] or *manipulation theorem* [Spirtes et al., 1993]. Appendix A details the standard graphical terminology.

**Independent Causal Mechanism (ICM)** postulates how Markov factors (hereon referred to as *causal mechanisms*) in Eq. (1) should relate to each other. Schölkopf et al. [2012] and Peters et al. [2017] express the insights as follows:

> Causal mechanisms are independent of each other in the sense that a change in one mechanism $P(X_i \mid \mathbf{PA}_i)$ does not inform or influence any of the other mechanisms $P(X_j \mid \mathbf{PA}_j)$, for $i \neq j$.

This notion of invariant, independent, and autonomous mechanisms has appeared in many forms throughout the history of causality research: from early work led by Haavelmo [1944] and Aldrich [1989] to Pearl [2009]. Studying properties of *independent causal mechanisms* rigorously demands a statistical understanding of what such independence means between distributions. Guo et al. [2023a] provides a statistical formalization of *ICM* in exchangeable data, thus providing necessary tools to study causal framework in exchangeable data. The next section introduces relevant background.

## 2.2 The Causal Framework in Exchangeable Data

**Definition 1 (Exchangeable Sequence)** *An exchangeable sequence of random variables is a finite or infinite sequence $X_1, X_2, X_3, \ldots$ such that for any finite permutation $\sigma$ of the position indices $\{1, \ldots, N\}$, the joint distribution of the permuted sequences remains unchanged to that of original:*

$$P(X_{\sigma(1)}, \ldots, X_{\sigma(N)}) = P(X_1, \ldots, X_N) \tag{3}$$

Note an *independent and identically distributed* (i.i.d.) sequence of random variables is an exchangeable sequence, i.e., $P(X_1, \ldots, X_N) = \prod_{i=1}^{N} P(X_i)$, but not all exchangeable sequences are i.i.d. Examples of exchangeable non-i.i.d. sequences include but are not limited to Pólya urn model [Hoppe, 1984], Chinese restaurant processes [Aldous et al., 1985] and Dirichlet processes [Ferguson, 1973].

An important result of exchangeable data are de Finetti's theorems [de Finetti, 1931] which show any exchangeable sequence can be represented as a mixture of conditionally i.i.d. data. Building upon the work of de Finetti [1931], Guo et al. [2023a] observes that exchangeable but not i.i.d. data possess extra conditional independence relationships compared to i.i.d. data. This enables:

**(a): Structural Causal Model**        **(b): ICM generative process**

| **Observational** | **do(X = x)** | | **Observational** | **do(X = x)** |
|---|---|---|---|---|
| $X := U_X$ | $X := x$ | | $\theta \sim p(\theta), \psi \sim p(\psi)$ | $X \sim \delta(X = x)$ |
| $Y := f(X, U_Y)$ | $Y := f(x, U_Y)$ | | $X \sim p(x|\theta), Y \sim p(y|x, \psi)$ | $Y \sim p(y|x, \psi)$ |

Figure 3: An illustration of differences in what the do-operator does between a structural causal model (a) and an ICM generative process (b). In the observational phase, SCMs (a), where the dotted plate indicates i.i.d. sampled, illustrates that fixed assignment of $U_X$ and $U_Y$ leads to fixed observable values $X$ and $Y$; on the other hand, for ICM-generative processes where *exch.* is an abbreviation for an exchangeable process, fixing $\theta, \psi$ does not fix $X$ and $Y$, instead, it means sampling from a fixed distribution. Because SCM fails to characterize ICM-generative processes, we define the operational meaning of do-interventions on ICM-generative processes as assigning $\delta$-distribution to the intervened variables and substituting the corresponding values in the remaining distributions.

- unique causal structure identification (Theorem 5 in [Guo et al., 2023a]), and
- statistical formalization of ICM principle (causal de Finetti theorems in [Guo et al., 2023a])

In contrast to previous work's focus on structure identification, this work studies causal effect identification and estimation in exchangeable data. Markovian models here mean there are no unmeasured confounders for each tuple of random variables observed in an exchangeable sequence.

To start with, we introduce the necessary terminologies inherited from Guo et al. [2023a].

**Definition 2 (ICM generative process)** *We say data is generated from an ICM generative process with respect to a DAG $\mathcal{G}$ if the sequence of random variable arrays $\{\mathbf{X}_{:;n}\}$ is infinitely exchangeable and satisfies $X_{i;[n]} \perp\!\!\!\perp \overline{\mathbf{ND}}_{i;[n]}, \mathbf{ND}_{i;n+1} | \mathbf{PA}_{i;[n]}$ for all $i \in [d]$ and $n \in \mathbb{N}$, where $\mathbf{PA}_i$ denotes parents of node $X_i$, $\mathbf{ND}_i$ denotes the corresponding non-descendants and $\overline{\mathbf{ND}}_i$ denotes the set of non-descendants excluding its own parents. By causal de Finetti theorems, it is equivalent to say the joint distribution of the sequence can be represented as:*

$$P(\mathbf{X}_{:;[N]} = \mathbf{x}_{:;[N]}) = \int \int \prod_{n=1}^{N} \prod_{i=1}^{d} p(x_{i;n} \mid \boldsymbol{pa}_{i;n}^{\mathcal{G}}, \boldsymbol{\theta_i}) d\nu_1(\boldsymbol{\theta_1}) \dots d\nu_d(\boldsymbol{\theta_d}), \tag{4}$$

*where $\nu_i$ are probability measures.*

In this work, we assume that any probability measure has a corresponding density function. Following Definition 2, an immediate result is any distribution $P$ generated by an ICM generative process is Markov to ICM($\mathcal{G}$). Definition 7 in Appendix A defines formally what we mean by an ICM operator on a DAG $\mathcal{G}$. Here we illustrate an example of ICM($\mathcal{G}$) and the use of our notation. Consider a DAG $\mathcal{G} := X_1 \leftarrow X_2 \rightarrow X_3$ with 3 variable indices. Two data tuples are generated under an ICM generative process with respect to $\mathcal{G}$. Fig. 2 shows the Markov structure compatible with the above ICM generative process.

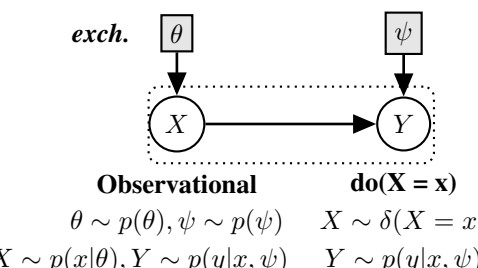

Figure 2: An example of ICM($\mathcal{G}$): Two data tuples are generated by an ICM generative process with respect to $\mathcal{G} := X_1 \leftarrow X_2 \rightarrow X_3$, where $X_{i;n}$ is the $i$-th variable in the $n$-th position and gray means latent variables.

## 3 Causal Effect in Exchangeable Data

We first motivate our study of causal effects in exchangeable data by noting exogenous variables are different from causal de Finetti parameters. Traditional causal effect in i.i.d. processes is defined as

graph surgery with respect to SCMs (cf. Section 2.1). Such a definition does not transfer its operational meaning to that in exchangeable processes because SCMs fail to characterize ICM generative processes. In this section, we first define what causal effect means and show their differences in properties compared to that of i.i.d. processes. We proceed to illustrate in the simplest possible case - a pair of random variables, and then provide a general statement for the multivariate case.

Figure 3 illustrates how SCMs fail to characterize ICM generative processes in the bivariate example. In SCMs, a fixed assignment of values to exogenous variables $U_X$ and $U_Y$ uniquely determines the values of all the observable variables $X$ and $Y$. This is not the case for ICM generative processes. A fixed assignment to causal de Finetti parameters $\theta$ and $\psi$ only restricts the observable variables $X$ and $Y$ to sample from fixed distributions but does not uniquely determine their values. Therefore, do-operator commonly defined as graph surgery in SCMs demands a new operational meaning in ICM generative processes. Definition 3 defines that $do(X = x)$ in ICM generative processes means assigning the sampling density $p(x|\theta)$ to $\delta(X = x)$.[2]. Doing so, we have:

$$\textbf{i.i.d. generative process}: P(Y = y|do(X = x)) = p(y|x, \psi_0) = P(Y = y|x), \psi_0 \in \mathbb{R} \qquad (5)$$

$$\textbf{ICM generative process}: P(Y = y|do(X = x)) = \int p(y|x, \psi)p(\psi)d\psi = P(Y = y|x) \qquad (6)$$

Despite being identical in expression, causal effect has different implications in i.i.d. and ICM generative processes. Under i.i.d., the randomness captured in $P$ is driven only by the randomness in exogenous variables $U_Y$. Under the ICM generative process, the randomness in $P$ is driven both by $p(y|x, \psi)$ and the randomness in the causal de Finetti parameter $p(\psi)$. It is well-known that i.i.d is a subcase of exchangeability. Here we observe that the causal effect expression follows the same pattern, i.e., the causal effect expression in i.i.d. is also a subcase of that under exchangeability whenever $p(\psi) = \delta(\psi = \psi_0)$.

Equipped with this operational meaning of intervention, we next consider the set of feasible intervention targets. Here we first clarify what we mean by the data-generating process. Data generated from i.i.d. or ICM processes refers to a sequence of random variables. For example, in the bivariate case, the sequence is $(X_1, Y_1), (X_2, Y_2), \ldots$. Often one omits position indices in i.i.d. because:

$$\textbf{i.i.d. generative process}: P(X_1, Y_1, \ldots, X_N, Y_N) \stackrel{ind}{=\!=} \prod_{n=1}^{N} P(X_n, Y_n) \stackrel{ide}{=\!=} [P(X, Y)]^N \qquad (7)$$

The first equality is due to independence and the second equality is due to identical distributions. One thus does not differentiate the position indices in i.i.d. as $P(X, Y)$ characterizes the joint distribution fully by Eq. (7). Intervention is thus defined only on $X$ and $Y$ rather than $X_n$ and $Y_n$. However, in the ICM generative process, $P(X, Y)$ cannot fully characterize the joint distribution. An example application of Definition 2 in the bivariate case gives:

$$\textbf{ICM gen. process}: P(x_1, y_1, \ldots, x_N, y_N) = \int \int \prod_{n=1}^{N} p(y_n|x_n, \psi)p(x_n|\theta)d\mu(\theta)d\nu(\psi) \qquad (8)$$

Eq. (8) shows that one can no longer omit the position indices in ICM generative processes, as $P(X, Y)$ cannot fully characterize the joint distribution. Intervention in ICM generative processes thus should be considered at the level of $X_n$ and $Y_n$. Definition 3, presented below, formalizes the concept of causal effect in ICM generative processes. In particular, both intervention sets and target variables of interest can be any random variables observed in the sequence.

**Definition 3 (Causal Effect in ICM generative processes)** *Let* $\mathbf{X}$ *and* $\mathbf{Y}$ *be two disjoint sets of variables generated by an ICM-generative process. For each realization* $\mathbf{x}$ *of* $\mathbf{X}$*,* $P(\mathbf{y}|do(\mathbf{x}))$ *in the ICM-generative process denotes the probability of* $\mathbf{Y} = \mathbf{y}$ *induced by assigning* $p(x_{i;n}|\mathbf{pa}_{i;n}^{\mathcal{G}}, \theta_i) = \delta(X_{i;n} = x_{i;n})$ *in Eq. (4),* $\forall X_{i;n} \in \mathbf{X}$*, and substituting* $\mathbf{X} = \mathbf{x}$ *in the remaining conditional distributions.*

Although marginal distribution $P(X, Y)$ cannot characterize the joint distribution in Eq. (8) due to mutual dependence of $X_n, Y_n$ with causal de Finetti parameters $\theta$ and $\psi$, we note variables in

---

[2]`https://encyclopediaofmath.org/index.php?title=Delta-function`

different positions share identical marginal distributions. Mathematically, data generated under an exchangeable process shares identical marginal distributions: $P(X_n, Y_n) = P(X_m, Y_m), \forall n \neq m$. This also means that identical interventions performed on variables in different positions result in the same post-interventional distributions: $P(Y_n|do(X_n = x)) = P(Y_m|do(X_m = x)), \forall n \neq m$. Similar to i.i.d., quantities such as $P(Y|do(X = x))$ are thus well-defined in ICM generative processes. Corollary 1 proves the multivariate equivalent of identical marginal intervention effects in ICM generative processes.

**Corollary 1 (Identical marginal post-interventional distributions)** *Let $P$ be the distribution for some ICM generative process. Let $\mathbf{I}$ and $\mathbf{J}$ be two disjoint subsets in $[d] := \{1, \ldots, d\}$. Denote $\mathbf{X}_{\mathbf{I};n} := \{X_{i;n} : i \in \mathbf{I}\}$ and similarly for $\mathbf{X}_{\mathbf{J};n}$. Then,*

$$P(\mathbf{X}_{J;n} \mid do(\mathbf{X}_{I;n} = \mathbf{x})) = P(\mathbf{X}_{J;m} \mid do(\mathbf{X}_{I;m} = \mathbf{x})), \forall n \neq m \quad (9)$$

*i.e., identical interventions on variables in different positions share the same marginal post-interventional distributions. See Appendix B for the proof.*

One can thus omit the notation of position indices when appropriate as supported by Corollary 1 and as illustrated in Figure 1 Block **A**. In the next section, we address a focal problem in causal effect estimation: causal effect identifiability in ICM generative processes.

## 3.1 Causal Effect Identifiability in ICM generative processes

Continuing with the bivariate example in Fig. 3 (b), consider $X_1, Y_1, X_2, Y_2$ is generated under an ICM generative process with respect to $\mathcal{G} := X \rightarrow Y$. Suppose one performs hard intervention on $X_1$, i.e., $do(X_1 = \hat{x})$. Applying traditional truncated factorization developed for i.i.d. data (Eq. (2)) yields:

$$P(x_1, y_1, x_2, y_2|do(X_1 = \hat{x})) = P(y_1|\hat{x})P(y_2|x_2)P(x_2)\mathbb{1}_{x_1 = \hat{x}} \quad (10)$$

However, the independence between $(X_1, Y_1)$ and $(X_2, Y_2)$ in i.i.d. generative processes does not hold in ICM generative processes. In fact, in ICM generative processes, applying Definition 3 gives:

$$P(x_1, y_1, x_2, y_2|do(X_1 = \hat{x})) = \int p(y_1|\hat{x}, \psi)p(y_2|x_2, \psi)p(\psi)d\psi p(x_2)\mathbb{1}_{x_1 = \hat{x}} \quad (11)$$

Eq. (11) does not equal to Eq. (10) whenever $p(\psi) \neq \delta(\psi = \psi_0)$ for some $\psi_0$. Thus, for data generated under ICM generative processes, a new theorem for truncated factorization is required. See Theorem 1 for the statement for general multivariate distributions and Appendix C for a detailed derivation.

**Theorem 1 (Truncated Factorization in ICM generative processes)** *For a given graph $\mathcal{G}$, let $P$ be the probability distribution for data generated under an ICM generative process with respect to $\mathcal{G}$ and let $p$ be the corresponding density. The post-interventional distribution after intervening on $\mathbf{X} = \hat{\mathbf{x}}$ has density given by:*

$$p(\mathbf{x}_{:;1}, \ldots, \mathbf{x}_{:;N}|do(\mathbf{X} = \hat{\mathbf{x}})) = \prod_{i \in I_{\mathbf{X}}} p(\mathbf{x}_{i;[\neg \mathbf{N}_i]}|\boldsymbol{pa}_{i;[\neg \mathbf{N}_i]}^{\mathcal{G}}) \prod_{i \notin I_{\mathbf{X}}} p(\mathbf{x}_{i;[N]}|\boldsymbol{pa}_{i;[N]}^{\mathcal{G}})\big|_{\mathbf{X} = \hat{\mathbf{x}}}, \quad (12)$$

*where $\mathbf{I}_{\mathbf{X}} := \{i : X_{i;n} \in \mathbf{X}\}$ denotes the set of variable indices being intervened on and $\mathbf{N}_i := \{n : X_{i;n} \in \mathbf{X}\}$ denotes the set of position indices corresponding to variable index $i$ in the intervention set $\mathbf{X}$ and $[\neg \mathbf{N}_i]$ denotes the set of positive integers less than or equal to $N$ excluding values in $\mathbf{N}_i$.*

Theorem 1 presents a procedure for computing the joint post-interventional distribution using pre-interventional conditional distributions when intervening on any set of variables in the Markovian model. This demonstrates that causal effects are identifiable in Markovian models under ICM generative processes. Note that the traditional truncated factorization in i.i.d. processes is again a special case of Eq. (12) just as i.i.d. processes are a special case of exchangeable processes.

## 3.2 Conditional Interventional distributions

The fundamental difference between i.i.d. and ICM generative processes lies in the violation of the independence condition inherent to i.i.d.. Consequently, interventional distributions computed by conditioning on observations differ between the two. Specifically, causal effect of $do(X_1 = \hat{x})$ on $Y_1$

given $X_2, Y_2$ differs when computed under i.i.d. (Eq. (13)) and ICM generative processes (Eq. (14)). This difference arises because $(X_1, Y_1) \perp\!\!\!\perp (X_2, Y_2)$ holds in i.i.d but not in ICM generative processes.

$$\textbf{i.i.d. generative processes} : P(Y_1|do(X_1 = \hat{x}), X_2, Y_2) = P(Y_1|\hat{x}) \tag{13}$$

$$\textbf{ICM generative processes} : P(Y_1|do(X_1 = \hat{x}), X_2, Y_2) = P(Y_1|\hat{x}, X_2, Y_2) \tag{14}$$

Fig. 1 (c) depicts this example of intervention via graph surgery on ICM($\mathcal{G}$) where the operational meaning of edge deletion is explained in Figure 3. We observe that conditioning on the collider node $Y_2$ in the ICM generative process renders $Y_1 \not\perp\!\!\!\perp X_2|Y_2$. Lemma 1 provides the corresponding general statement for the multivariate case when conditioning on other observations. See Appendix D for the proof.

**Lemma 1 (Intervention effect conditioned on other observations)** *For a given graph $\mathcal{G}$, let $P$ be the distribution for the ICM generative process with respect to $\mathcal{G}$. Let $\mathbf{X}$ be the intervention set. Assume $\mathbf{X} = \mathbf{X}_{\mathbf{I};n} := \{X_{i;n} : \forall i \in \mathbf{I}\}$. Let $\mathbf{S} \subseteq [N]$ such that $n \notin \mathbf{S}$ and $[\neg\mathbf{I}]$ denotes $[d]\backslash\mathbf{I}$. Then,*

$$P(\mathbf{X}_{\neg\mathbf{I};n}|do(\mathbf{X}_{\mathbf{I};n} = \hat{\mathbf{x}}), \mathbf{X}_{:;\mathbf{S}}) = \prod_{i \notin \mathbf{I}} P(\mathbf{X}_{i;n}|\mathbf{X}_{i;\mathbf{S}}, \boldsymbol{PA}_{i;\mathbf{S} \cup \{n\}})|_{\mathbf{X}_{\mathbf{I};n}=\hat{\mathbf{x}}} \tag{15}$$

A similar argument also applies to the intervention effect conditioned on observations of experimental results performed on other tuples of random variables in the sequence. Appendix E discusses in detail. Here we show the implications of conditional interventional distributions via a causal Pólya urn model.

**Causal Pólya Urn Model** Imagine an urn with left and right compartments. The experimenter puts $\alpha$ white balls and $\beta$ black balls in each compartment. At each step $n$, one ball is uniformly drawn from the left and one ball is uniformly drawn from the right. The chosen two balls in the order of left and right are then placed in a dark chamber unobserved by the experimenter. A hidden mechanism reads the color of the two balls and outputs $X_n, Y_n$ to the experimenter. The mechanism outputs $X_n = 1$ whenever the $n$-th left ball is black else $X_n = 0$ and $Y_n = 1$ whenever the left and right balls disagree in color. After observing $X_n$ and $Y_n$, the experimenter puts the original balls back in the corresponding compartment and add a ball of the same color as $X_n$ in the left and add a ball of the same color as $Z_n := (1 - X_n) * Y_n + (1 - Y_n) * X_n$ in the right.

**Causal de Finetti application** The causal Pólya urn model is a real-world illustration of causal de Finetti thereom in its bivariate form. The joint distribution of observed sequence $P(x_1, y_1, \ldots, x_n, y_n)$ for all $n \in \mathbb{N}$ can be perfectly modelled as the RHS of Eq. (4) with two variables and $X \rightarrow Y$:

$$\int \int \prod_n p(y_n \mid x_n, \psi)p(x_n \mid \theta)p(\theta)p(\psi)d\theta d\psi,$$

where $p(\theta), p(\psi)$ are Beta distributions and $p(y_n \mid x_n, \psi), p(x_n \mid \theta)$ are Bernoulli distributions. Appendix F provides a detailed analysis of how *causal Pólya Urn Model* satisfies both exchangeability and certain conditional independences conditions. Therefore causal de Finetti theorems apply to the causal Pólya urn model in the sense the joint observational distribution can be equivalently modelled as ICM generative processes in bivariate form. Next, we study how intervention propagates in this model.

**Discussion** For a given $n$, if $X_m = 1, Y_m = 0$ for many $m < n$, then it is more likely $X_n = 1, Y_n = 0$ as there are more black balls are added to both left and right urn compartments. Consider an intervention: imagine an agent replaces the left ball in the dark chamber to be a white ball for the $n$-th draw, $do(X_n = 0)$, then if $X_m = 1, Y_m = 0$ for many $m < n$, it is more likely that $Y_n = 1$ as more black balls are placed in the right compartment and $X_n$ is fixed to be a white ball by intervention. This illustrates a practical example of the aforementioned conditional post-interventional distributions. After the intervention, the observer, ignorant of what happens in the dark chamber, proceeds as normal with the replacement process. The intervention thus changes the dynamics of causal Pólya urn model. We consider causal Pólya model interesting as it illustrates through simple game-like settings that observables $(X_n, Y_n)$ satisfying $Y_{[n]} \perp\!\!\!\perp X_{n+1}|X_{[n]}$ are indeed driven by the independent mechanisms hidden from the observers. This is akin to how Nature hides causal mechanisms from observers and scientists can only reason through observables.

### 3.3 Rules of compact representation of causal effect

From Theorem 1, any interventional effect $P(\mathbf{Y}|do(\mathbf{X}))$ can be obtained from marginalization. However, in practice, marginalization on many variables is computationally expensive. Further, Theorem

1 requires observations and measurements of all joint variables which is resource-intensive in practice. Such problem on *the observability of variables* is more explicitly posed as out-of-variable problem in Guo et al. [2023b]. In this section, we present rules that allow simplification of causal expressions.

**Lemma 2 (Intervention effect on differing positions)** *Given a graph $\mathcal{G}$ and let $P$ be the distribution for the ICM generative process with respect to $\mathcal{G}$. Let $\mathbf{X}$ and $\mathbf{Y}$ be two disjoint sets such that $\mathbf{X}$ is the intervention set and $\mathbf{Y}$ is the target set. Let $\mathbf{N_X} := \{n : X_{i;n} \in \mathbf{X}\}$ be the set of position indices being intervened, and similarly $\mathbf{N_Y}$ be the set of position indices being targeted to observe. Assume $\mathbf{N_X} \cap \mathbf{N_Y} = \emptyset$. Then,*

$$P(\mathbf{Y} \mid do(\mathbf{X} = \mathbf{x})) = P(\mathbf{Y}), \tag{16}$$

**Lemma 3 (Intervention effect within the same position)** *Given a graph $\mathcal{G}$ and let $P$ be the distribution for the ICM generative process with respect to $\mathcal{G}$. Let $\mathbf{X}$ be the intervention set such that it consists only $\mathbf{X}_{\mathbf{I};n}$ where $n \in \mathbf{S} \subseteq [N]$ and $\mathbf{I} \subseteq [d]$ is a set of variable indices. Let $\mathbf{Y}$ be the target set such that it consists only $\mathbf{X}_{\mathbf{J};n}$ where $n \in \mathbf{S} \subseteq [N]$ and $\mathbf{J} \subseteq [d]$. Note $\mathbf{I} \cap \mathbf{J} = \emptyset$. Then,*

$$P(\mathbf{Y} \mid do(\mathbf{X} = \hat{\mathbf{x}})) = \sum_{\boldsymbol{pa_X}} P(\mathbf{Y} \mid \hat{\mathbf{x}}, \boldsymbol{pa_X}) P(\boldsymbol{pa_X}), \tag{17}$$

*where $\boldsymbol{PA_X}$ denotes the parent set of intervened variables $\mathbf{X}$.*

Lemma 2 shows that the interventional distributions on target variables of interest are unchanged when only intervening on variables in differing positions other than the target variables. Lemma 3 shows for arbitrary causal queries when acting on a consistent set of variables across positions (i.e., the intervention set $\mathbf{X}$ consists of $\mathbf{X}_{\mathbf{I};n}$, where each position $n$ shares the same set of variable indices $\mathbf{I}$ being intervened on), the post-interventional distribution can be estimated by only observing and measuring $\mathbf{Y}, \mathbf{X}, \mathbf{PA_X}$, where $\mathbf{PA_X}$ denotes the parent set of $\mathbf{X}$. This is consistent with the parent adjustment formula under i.i.d. generative processes, with the additional dependence structure across variables in differing positions. See Appendix G and H for detailed statements and proofs.

## 4  Causal Effect in Multi-environment data

In Section 3, we establish the identifiability of causal effects given the graph $\mathcal{G}$ and the distribution $P$ generated by an ICM generative process. Note under i.i.d. generative processes, causal effect identification hinges on the knowledge of causal diagram $\mathcal{G}$. Here, we show that generalizing causality to an exchangeable framework does not necessarily mean we have less ability to perform graphical identification and effect estimation. In fact, with an unknown graph, ICM generative processes allow one to identify graph and causal effects simultaneously. In other words, data adhering to the ICM principle and generated under an exchangeable process alone is sufficient for both graphical and effect identification.

**Theorem 2 (Causal effect identification in ICM generative processes)** *Denote $\mathbf{Y}, \mathbf{X}$ be two disjoint subsets of observable variables in $\{\mathbf{X}_{:;n}\}_{n \in \mathbb{N}}$. Then $P(\mathbf{Y}|do(\mathbf{X} = \mathbf{x}))$ is identifiable given the distribution $P$ from the class of distributions generated from ICM generative processes. Here identifiability means the causal query can be computed uniquely from $P$.*

Informally, Theorem 2 says that for ICM generative processes, both causal graphs and causal effects can be identified simultaneously. This is derived from a concatenation of Theorem 5 in Guo et al. [2023a] and Theorem 1 in the current work. Theorem 5 in Guo et al. [2023a] shows for ICM generative processes, causal graph is identifiable and Theorem 1 shows the computation, and thus identifiability, of causal effects in ICM generative processes given graph.

To see how to apply Theorem 2 in practice, we build connection between exchangeable and grouped or multi-environment data and propose the *Do-Finetti* algorithm. In the causal literature, grouped data refers to data available from multiple environments, each producing (conditionally) i. i. d observations from a different distribution, which are related through some invariant causal structure shared by all environments. We can interpret multi-environment data through the lens of exchangeability as follow: In each environment $e$, we observe exchangeable random variables $\mathbf{X}^e_{:;[N_e]}$, where $X^e_{d;n}$ denotes the $d$-th random variable observed at $n$-th position in environment $e$. $N_e$ denotes the number of observations

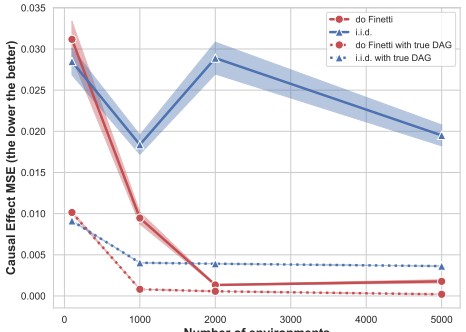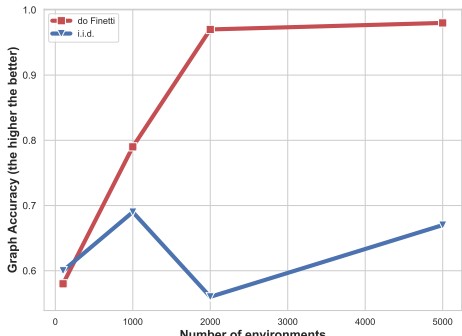

Figure 4: Our method's (*do-Finetti*) performance in simultaneously identifying DAG (right) and causal effect estimation (left), compared to the i.i.d. algorithm (*i.i.d.*) and corresponding methods with known true DAG (*do Finetti with true DAG* and *i.i.d. with true DAG*) in bivariate setting. Left shown are the mean and standard deviation of MSE compared to analytic solutions for each method aggregated over 100 experiments. Right shows the accuracy of identifying the correct underlying DAG for each method. *Do-Finetti* identifies unique causal structures and achieves near-perfect causal effect estimation.

from environment $e$. Data across environments are independent and identically distributed in the sense that the distribution of $\mathbf{X}^e_{:;[N]}$ and $\mathbf{X}^{e'}_{:;[N]}$ is identical for all $N < \min(N_e, N_{e'})$. Each environment thus provides a finite marginal of an i. i.d copy of the same exchangeable process.

The *Do-Finetti* algorithm combines *Causal-de-Finetti* algorithm developed in Guo et al. [2023a] and the truncated factorization for ICM generative processes developed in Section 3 of this paper. It provides a proof-of-concept algorithm to verify that multi-environment data generated under the independent causal mechanism principle alone via an exchangeable process can simultaneously estimate the causal effect and causal graph. See Algorithm 1 in Appendix I for details of the procedure.

## 5   Experiments

We construct synthetic datasets according to causal Pólya urn model (cf. Section 3.2) and demonstrate that *Do-Finetti* algorithm can estimate causal effects and graphs simultaneously. We compare with the standard method in i.i.d. processes (i.e., PC algorithm and truncated factorization Eq. (2)) and show empirically that the traditional truncated factorization cannot estimate causal effects in our setting. We then perform ablation studies to attribute errors to either graphical misclassification or effect estimation.

Let superscript $^e$ denote variables generated in environment $e$. The data-generating process for $X \to Y$, for example, as follows:

$$\theta^e \sim \mathrm{Beta}(\alpha, \beta), \psi^e \sim \mathrm{Beta}(\alpha, \beta)$$
$$X \to Y : X_i^e := \mathrm{Ber}(\theta^e), Y_i^e := \mathrm{Ber}(\psi^e) \oplus X_i^e$$

where $\oplus$ denotes xor operation and $X_i^e, Y_i^e$ denote variables generated at the $i$-th position in environment $e$. We collect two pairs of random variables across all environments and run *do-Finetti* algorithm to compute the post-interventional distributions with randomly initialized intervened variable and its corresponding values as in Eq. (12). We repeat the experiment for 100 times and report the mean squared error loss between predicted and analytic solutions across varying number of environments. Fig. 4 shows i.i.d. fails to estimate causal effects, giving high estimation error in ICM generative processes. Even with knowledge of the true DAG and infinite data, *i.i.d. with true DAG* never achieves analytic solutions (Fig. 4a). On the other hand, *do-Finetti* achieves near-zero causal effect estimation error, meaning correct DAG identification and correct effect estimation. Appendix K details exact experiment setups.

# 6 Discussion

**Causality and exchangeability.** Causal effect estimation in randomized controlled trials [Rubin, 2005] relies on the exchangeability assumption between the controlled and treatment group. More recent work [Dawid, 2021] introduces a decision-theoretic framework for causality and uses pre-treatment and post-treatment exchangeability as foundational assumptions on external data used to solve a decision problem. Jensen et al. [2020] studies exchangeability in the context of object conditioning. It shows object conditioning, due to exchangeability, mitigates latent confounding and measurement errors for causal inference. Guo et al. [2023a] characterize independent causal mechanisms in exchangeable data via causal de Finetti theorems. Reizinger et al. [2024] proposes relaxed conditions for causal discovery in ICM generative processes and shows exchangeable non-i.i.d. data is the key to both structure and representation identifiability. This work extends on causal de Finetti and studies causal effect identification and estimation in ICM generative processes.

**Causal effect estimation in multi-environment data.** Bareinboim and Pearl [2016] study the transportability of causal effects for populations under different experimental conditions. Peters and Meinshausen [2016] quantify causal effect estimation and its uncertainties through exploiting invariance of causal mechanisms. Jaber et al. [2019] develop a complete algorithm for causal effect identification in the Markov equivalence class of causal graphs which was previously deemed impossible to identify further without additional assumptions in the i.i.d. framework. In the present work, we show ICM generative processes, naturally arising in multi-environment data, allow simultaneous causal structure recovery and effect identification. Further, we illustrate via a causal Pólya urn model that an exchangeable process, though related, is not limited to only multi-environment data.

**Causality in non-i.i.d. data.** A classical problem in causality is the study of *interference*, motivated by real-world complex interdependencies within subjects. For example, in social networks and infectious disease, treatments on one subject affects outcomes of other subjects. Sherman [2022] provides a general identification formula of causal estimands under dependent data when graphs contain unobserved confounders. Ogburn and VanderWeele [2014] provides a comprehensive review on causal inference under interference and studies how to determine what variables must be measured for computation of causal estimands. Zhang et al. [2023] provides an elegant study on modeling uncertain interaction using linear graphical causal models. Maier [2014] studies causal discovery in relational data. This work focuses on a particular type of non-i.i.d. data, namely exchangeable data. In contrast with classical setting where we assume aprior known causal graph and develop causal estimands from thereon, the series of *causal de Finetti* work focus on empirical inference of causality from exchangeable non-i.i.d. data. This we hope paves the way to uncover governing laws of Nature from empirical observations.

**Real-world Applications.** Exchangeable data offers an expressive and realistic representation of complex structured relational data, which often appears in clinical studies [Bowman and George, 1995], microarray gene expression data [Qin, 2006], or in high-dimensional inference tasks for images [Korshunova et al., 2018, 2020], 3D point cloud modeling [Yang et al., 2020] to topic modeling [McAuliffe and Blei, 2007]. While it is beyond the scope of the present work, we hope that our work, through building a theoretical framework of interventions in exchangeable data, may ultimately help study and understand naturally occurring mechanisms in scientific domains.

# 7 Conclusion

We study causal effects and prove a generalized truncated factorization for an important class of exchangeable generative processes. We show for conditional interventional distributions that other observations are relevant to the causal query for exchangeable but not for i.i.d. data. We introduce causal Pólya urn models and demonstrate in-practice interventions in exchangeable data. We develop a *Do-Finetti* algorithm that performs simultaneous DAG and effect estimation from multi-environment data.

It is exciting to start to understand the complexities of non-i.i.d. data, and much is left to do: from developing algorithms that perform at scale to understanding counterfactual queries in exchangeable contexts. Appendix L details limitations and broader impact. Going beyond i.i.d. has been a major bottleneck when applying machine learning to real-world applications, and exchangeable data offers a doable and realistic next step.

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

## Table of Contents

# A Graphical Terminology

A graph $\mathcal{G}$ consists of vertices $V$ and edges $E$. The set of vertices is often denoted as $\{X_1, \ldots, X_n\}$. We say a pair of vertices is connected with a directed edge if $X_i \to X_j$. The set of edges $E$ contains a set of pairs $\{X_i \to X_j : X_i, X_j \in V\}$. If there does not exist a sequence of edges such that $X_i \to \cdots \to X_i$ for some $X_i \in V$, then $\mathcal{G}$ is acyclic. Whenever $X_i \to X_j \in E$, then $X_j$ is the child of $X_i$ and that $X_i$ is the parent of $X_j$ in $\mathcal{G}$. One use $\mathbf{PA}_i$ to denote the parents of $X_i$ and $\mathbf{CH}_i$ to denote its children. A sequence of vertices $X_1, \ldots, X_k$ forms a directed path in $\mathcal{G}$ if, for every $i = 1, \ldots, k$, we have $X_i \to X_{i+1}$. One say that $X$ is an ancestor of $Y$ in $\mathcal{G}$, and that $Y$ is a descendant of $X$, if there exists a directed path $X_1, \ldots, X_k$ with $X_1 = X$ and $X_k = Y$. We write $\mathbf{DE}_i$ as descendants of $X_i$ and $\mathbf{ND}_i$ as non-descendants of $X_i$ and $\overline{\mathbf{ND}}_i$ as non-decendants of $X_i$ excluding its own parents.

Both graph and probability distribution encode conditional independence relationships. Given a probability distribution $P$, we say $X \perp\!\!\!\perp Y | W$, if $P(X|Y, W) = P(X|W)$. In graphical terms, we detect the set of conditional independence in a DAG through d-separation.

**Definition 4 (d-separation)** *A path $p$ is d-separated by a block of node $Z$ if and only if one of the two conditions holds: 1) $p$ contains a chain $X_i \to X_m \to X_j$ or a fork $X_i \leftarrow X_m \to X_j$ s.t. $X_m \in Z$; 2) $p$ contains $X_i \to X_m \leftarrow X_j$ s.t. the middle node $X_m \notin Z$ and $\boldsymbol{DE}_m \notin Z$. We then say $Z$ d-separates $X$ and $Y$ if it blocks every path from a node in $X$ to a node in $Y$.*

**Definition 5 (I-map)** *Given a probability distribution $P$, $\mathcal{I}(P)$ denotes the set of conditional independence relationships of the form $X \perp\!\!\!\perp Y \mid Z$ that hold in $P$. Given a DAG $\mathcal{G}$, $\mathcal{I}(\mathcal{G})$ denotes the set of conditional independence assumptions read-off via d-separation.*

**Definition 6 (Markovian and Faithful)** *A distribution $P$ is Markov to a DAG $G$ if $\mathcal{I}(G) \subseteq \mathcal{I}(P)$. A distribution $P$ is faithful to $G$ if $\mathcal{I}(P) \subseteq \mathcal{I}(G)$.*

Next we proceed to define what is ICM($\mathcal{G}$), through explicating the ICM operator.

**Definition 7 (ICM operator on a DAG $\mathcal{G}$)** *Let $U$ be the space of DAGs with vertices $X_1, \ldots, X_d$. Let $V$ be the space of acyclic directed mixed graphs (ADMG [Richardson, 2003]) with vertices $\{(X_{i;n})\}$, where $i \in [d], n \in \mathbb{N}$. A mapping $F$ from $U$ to $V$ is an ICM operator if $F(\mathcal{G})$ satisfies 1) $F(\mathcal{G})$ restricted to the subset of vertices $\{X_{1;n}, \ldots, X_{d;n}\}$ is a DAG $\mathcal{G}$, for any $n \in \mathbb{N}$; 2) $X_{i;n} \leftrightarrow X_{i;m}$ whenever $n \neq m$ for all $i \in [d]$; 3) there are no other edges other than those stated above. We denote the resulting ADMG as ICM($\mathcal{G}$) and $\boldsymbol{PA}_{i;n}^{\mathcal{G}}$ denotes the parents of $X_{i;n}$ and similarly $\boldsymbol{ND}_{i;n}^{\mathcal{G}}$ for the corresponding non-descendants.*

# B Proof of Corollary 1

**Definition 8 (Exchangeable arrays)** *An array of size $d$ contains variables $(X_{1;n}, \ldots, X_{d;n})$ where $X_{d;n}$ denotes the $d$-th random variable in $n$-th array. Such an array is denoted as $\mathbf{X}_{:;n}$. A finite sequence of size $d$ arrays is **exchangeable**, if for any permutation map $\pi$,*

$$P(\mathbf{X}_{:;\pi(1)}, \ldots, \mathbf{X}_{:;\pi(N)}) = P(\mathbf{X}_{:;1}, \ldots, \mathbf{X}_{:;N}) \tag{18}$$

Next we show exchangeability implies identical marginal distributions. This is a standard result in exchangeable sequences [de Finetti, 1931], here we include it for completeness.

**Theorem 3 (Exchangeable implies identical marginal distribution)** *Let $P$ be the distribution for some exchangeable processes, where $\mathbf{X}_{:;1}, \ldots, \mathbf{X}_{:;N}$ are exchangeable arrays. Then for any $n \neq m$:*

$$P(\mathbf{X}_{:;n}) = P(\mathbf{X}_{:;m}) \tag{19}$$

**Proof of Theorem 3** *Without loss of generality, assume $n < m$. Let $\pi : [N] \to [N]$ be the permutation mapping such that $\pi(i) = i, \forall i \neq \{n, m\}$ and $\pi(n) = m, \pi(m) = n$.*

*Then, by definition of exchangeability,*

$$P(\mathbf{X}_{:;1} = \mathbf{x}_{:;1}, \ldots, \mathbf{X}_{:;n} = \mathbf{x}_{:;n}, \ldots, \mathbf{X}_{:;m} = \mathbf{x}_{:;m}, \ldots, \mathbf{X}_{:;N} = \mathbf{x}_{:;N}) \tag{20}$$

$$= P(\mathbf{X}_{:;1} = \mathbf{x}_{:;1}, \ldots, \mathbf{X}_{:;n} = \mathbf{x}_{:;m}, \ldots, \mathbf{X}_{:;m} = \mathbf{x}_{:;n}, \ldots, \mathbf{X}_{:;N} = \mathbf{x}_{:;N}) \tag{21}$$

*Integrating out every values $\mathbf{x}_{:;m}, m \neq n$, we have $P(\mathbf{X}_{:;m} = \mathbf{x}_{:;n}) = P(\mathbf{X}_{:;n} = \mathbf{x}_{:;n})$ for all values $\mathbf{x}_{:;n}$. Therefore, $\mathbf{X}_{:;m}$ and $\mathbf{X}_{:;n}$ shares identical marginal distribution.*

**Corollary 1 (Identical marginal post-interventional distributions)** *Let $P$ be the distribution for some ICM generative process. Let $\mathbf{I}$ and $\mathbf{J}$ be two disjoint subsets in $[d] := \{1, \ldots, d\}$. Denote $\mathbf{X}_{\mathbf{I};n} := \{X_{i;n} : i \in \mathbf{I}\}$ and similarly for $\mathbf{X}_{\mathbf{J};n}$. Then,*

$$P(\mathbf{X}_{J;n} \mid do(\mathbf{X}_{I;n} = \mathbf{x})) = P(\mathbf{X}_{J;m} \mid do(\mathbf{X}_{I;m} = \mathbf{x})), \forall n \neq m \quad (9)$$

*i.e., identical interventions on variables in different positions share the same marginal post-interventional distributions. See Appendix B for the proof.*

**Proof of Corollary 1** *By Theorem 3, $P(\mathbf{X}_{:;n}) = P(\mathbf{X}_{:;m}), \forall n \neq m$. One can thus drop the notation of position indices when considering $\mathbf{X}_{:;n}$ for some $n$ as $\mathbf{X}_{:;n} \sim P$ from identical $P$. Due to truncated factorization in i.i.d. case (Eq. 2), any post-interventional distributions can be represented as pre-interventional distributions. Thus post-interventional distributions in $P(\mathbf{X}_{:;n} = \mathbf{x}_{:;n} | do(\mathbf{X}_{I;n} = \mathbf{x}))$ is uniquely determined by $P$ and so is $P(\mathbf{X}_{:;m} = \mathbf{x}_{:;n} | do(\mathbf{X}_{I;m} = \mathbf{x}))$. Since $P$ is identical, then so is the post-interventional distributions. Marginalizing out irrelevant variable indices that are not contained in $\mathbf{I} \cup \mathbf{J}$ gives the desired result.*

## C Proof of Theorem 1

**Notation** Let $X_{i;n}$ denote the random variable corresponding to $i$-th variable index and $n$-th position index for data generated under ICM generative processes. Write $\mathbf{X}_{:;n} := (X_{1;n}, \ldots, X_{d;n})$ and $\mathbf{X}_{d;[N]} := (X_{d;1}, \ldots, X_{d;N})$. Define $\mathbf{UP}_{i;:} := (\mathbf{X}_{i+1;:}, \ldots, \mathbf{X}_{d;:})$, which contains all random variables that have higher variable index value than $i$, i.e. upstream of node $i$.

**Theorem 1 (Truncated Factorization in ICM generative processes)** *For a given graph $\mathcal{G}$, let $P$ be the probability distribution for data generated under an ICM generative process with respect to $\mathcal{G}$ and let $p$ be the corresponding density. The post-interventional distribution after intervening on $\mathbf{X} = \hat{\mathbf{x}}$ has density given by:*

$$p(\mathbf{x}_{:;1}, \ldots, \mathbf{x}_{:;N} | do(\mathbf{X} = \hat{\mathbf{x}})) = \prod_{i \in I_{\mathbf{X}}} p(\mathbf{x}_{i;[\neg\mathbf{N}_i]} | \boldsymbol{pa}_{i;[\neg\mathbf{N}_i]}^{\mathcal{G}}) \prod_{i \notin I_{\mathbf{X}}} p(\mathbf{x}_{i;[N]} | \boldsymbol{pa}_{i;[N]}^{\mathcal{G}}) \big|_{\mathbf{X}=\hat{\mathbf{x}}}, \quad (12)$$

*where $\mathbf{I}_{\mathbf{X}} := \{i : X_{i;n} \in \mathbf{X}\}$ denotes the set of variable indices being intervened on and $\mathbf{N}_i := \{n : X_{i;n} \in \mathbf{X}\}$ denotes the set of position indices corresponding to variable index $i$ in the intervention set $\mathbf{X}$ and $[\neg\mathbf{N}_i]$ denotes the set of positive integers less than or equal to $N$ excluding values in $\mathbf{N}_i$.*

**Proof of Theorem 1** *Without loss of generality, we reorder the variables according to reversed topological ordering, i.e., a node's parents will be placed after this node. Note a reversed topological ordering is not unique, but it must satisfy a node's descendants will come before itself.*

*By chain rule, $P(\mathbf{X}_{:;1}, \ldots, \mathbf{X}_{:;N}) = \prod_i P(\mathbf{X}_{i;[N]} | \mathbf{UP}_{i;[N]})$. By Theorem 7 (Causal conditional de Finetti) in Guo et al. [2023a], since $P$ is generated under some ICM generative process, there exists a latent variable $\boldsymbol{\theta}_i$ with suitable probability measure $\nu$ such that*

$$P(\mathbf{x}_{i;[N]} | \boldsymbol{up}_{i;[N]}) = \int \prod_{n=1}^{N} p(x_{i;n} | \boldsymbol{pa}_{i;n}^{\mathcal{G}}, \boldsymbol{\theta}_i) d\nu(\boldsymbol{\theta}_i) \quad (22)$$

$$= \int \prod_{n=1}^{N} p(x_{i;n} | \boldsymbol{pa}_{i;n}^{\mathcal{G}}, \boldsymbol{\theta}_i) p(\boldsymbol{\theta}_i) d\boldsymbol{\theta}_i \quad (23)$$

*In this work, assume for every probability measure $\nu$ there exists a corresponding probability density associated with it and thus the second equality follows.*

*Note by the second condition in causal de Finetti theorem - multivariate version in Guo et al. [2023a], we have*

$$X_{i;[n]} \perp\!\!\!\perp \overline{\boldsymbol{ND}}_{i;[n]} | \boldsymbol{PA}_{i;[n]}$$

*where $\boldsymbol{PA}_i$ selects parents of node $i$ and $\boldsymbol{ND}_i$ selects non-descendants of node $i$ in $\mathcal{G}$. $\overline{\boldsymbol{ND}}_i$ denotes the set of non-descendants of node $i$ excluding its own parents. Then the naive application of the conditional independence on $P(\mathbf{X}_{i;[N]} | \mathbf{UP}_{i;[N]})$ gives:*

$$P(\mathbf{X}_{i;[N]} | \mathbf{UP}_{i;[N]}) = P(\mathbf{X}_{i;[N]} | \mathbf{PA}_{i;[N]})$$

*Combining gives:*

$$P(\mathbf{x}_{i;[N]}|\boldsymbol{up}_{i;[N]}) = P(\mathbf{x}_{i;[N]}|\boldsymbol{pa}_{i;[N]}^{\mathcal{G}}) = \int \prod_{n=1}^{N} p(x_{i;n}|\boldsymbol{pa}_{i;n}^{\mathcal{G}}, \boldsymbol{\theta}_i)p(\boldsymbol{\theta}_i)d\boldsymbol{\theta}_i \tag{24}$$

*Applying Eq. 24 on the chain rule decomposition of the joint distribution,*

$$P(\mathbf{X}_{:;1}, \ldots, \mathbf{X}_{:;N}) = \prod_i P(\mathbf{X}_{i;[N]}|\boldsymbol{PA}_{i;[N]}^{\mathcal{G}}) \tag{25}$$

*Inheriting Definition 3:*

*When $i \notin I_{\mathbf{X}}$, intervention on $\mathbf{X}$ does not change the probability distribution but just enforces consistency $\boldsymbol{PA}_{i;n}^{\mathcal{G}}$ with values in $\hat{\mathbf{x}}$, i.e.,*

$$P(\mathbf{X}_{i;[N]}|\boldsymbol{PA}_{i;[N]}^{\mathcal{G}}, do(\mathbf{X} = \hat{\mathbf{x}})) = P(\mathbf{X}_{i;[N]}|\boldsymbol{PA}_{i;[N]}^{\mathcal{G}})\big|_{\mathbf{X}=\hat{\mathbf{x}}}, \quad \forall i \notin I_{\mathbf{X}} \tag{26}$$

*When $i \in I_{\mathbf{X}}$, intervention on $\mathbf{X}$ affect variables only contained in the subset of $\mathbf{X}_{i;\mathbf{N}_i}$. Let $X_{i;n} \in \mathbf{X}_{i;\mathbf{N}_i}$. Then $p(x_{i;n}|\boldsymbol{pa}_{i;n}^{\mathcal{G}}, \boldsymbol{\theta}_i) = \delta(X_{i;n} = \hat{x}_{i;n})$. Aggregating together,*

$$P(\mathbf{x}_{i;[N]}|\boldsymbol{pa}_{i;[N]}^{\mathcal{G}}, do(\mathbf{X} = \hat{\mathbf{x}})) = \int \prod_{n \in \mathbf{N}_i} \delta(X_{i;n} = \hat{x}_{i;n}) \prod_{n \in [\neg\mathbf{N}_i]} p(x_{i;n}|\boldsymbol{pa}_{i;n}^{\mathcal{G}}, \boldsymbol{\theta}_i)p(\boldsymbol{\theta}_i)d\boldsymbol{\theta}_i \tag{27}$$

$$= \begin{cases} P(\mathbf{x}_{i;[\neg\mathbf{N}_i]}|\boldsymbol{pa}_{i;[\neg\mathbf{N}_i]}^{\mathcal{G}}) & \text{when } \boldsymbol{pa}_{i;n} \text{ consistent with } \mathbf{X}, \forall n \in \mathbf{N}_i \\ 0 & \text{else} \end{cases} \tag{28}$$

$$= P(\mathbf{x}_{i;[\neg N_i]}|\boldsymbol{pa}_{i;[\neg N_i]}^{\mathcal{G}})\big|_{\mathbf{X}=\hat{\mathbf{x}}}, \quad \forall i \in I_{\mathbf{X}} \tag{29}$$

*Combining Eq. 26 and 29,*

$$P(\mathbf{X}_{:;1}, \ldots, \mathbf{X}_{:;N}|do(\mathbf{X} = \hat{\mathbf{x}})) = \prod_{i \in I_{\mathbf{X}}} P(\mathbf{X}_{i;[\neg\mathbf{N}_i]}|\boldsymbol{PA}_{i;[\neg\mathbf{N}_i]}^{\mathcal{G}}) \prod_{i \notin I_{\mathbf{X}}} P(\mathbf{X}_{i;[N]}|\boldsymbol{PA}_{i;[N]}^{\mathcal{G}})\big|_{\mathbf{X}=\hat{\mathbf{x}}} \tag{30}$$

# D   Proof of Lemma 1

**Lemma 1 (Intervention effect conditioned on other observations)** *For a given graph $\mathcal{G}$, let $P$ be the distribution for the ICM generative process with respect to $\mathcal{G}$. Let $\mathbf{X}$ be the intervention set. Assume $\mathbf{X} = \mathbf{X}_{\mathbf{I};n} := \{X_{i;n} : \forall i \in \mathbf{I}\}$. Let $\mathbf{S} \subseteq [N]$ such that $n \notin \mathbf{S}$ and $[\neg\mathbf{I}]$ denotes $[d]\backslash\mathbf{I}$. Then,*

$$P(\mathbf{X}_{\neg\mathbf{I};n}|do(\mathbf{X}_{\mathbf{I};n} = \hat{\mathbf{x}}), \mathbf{X}_{:;\mathbf{S}}) = \prod_{i \notin \mathbf{I}} P(\mathbf{X}_{i;n}|\mathbf{X}_{i;\mathbf{S}}, \boldsymbol{PA}_{i;\mathbf{S}\cup\{n\}})|_{\mathbf{X}_{\mathbf{I};n}=\hat{\mathbf{x}}} \tag{15}$$

**Proof of Lemma 1** *By Theorem 1,*

$$P(\mathbf{X}_{:;\mathbf{S}\cup\{n\}}|do(\mathbf{X}_{I;n} = \hat{\mathbf{x}})) = \prod_{i \in I} P(\mathbf{X}_{i;\mathbf{S}}|\boldsymbol{PA}_{i;\mathbf{S}}^{\mathcal{G}}) \prod_{i \notin I} P(\mathbf{X}_{i;\mathbf{S}\cup\{n\}}|\boldsymbol{PA}_{i;\mathbf{S}\cup\{n\}}^{\mathcal{G}})\big|_{\mathbf{X}_{I;n}=\hat{\mathbf{x}}} \tag{31}$$

*By Lemma 2, since $n \notin \mathbf{S}$,*

$$P(\mathbf{X}_{:;\mathbf{S}}|do(\mathbf{X}_{I;n} = \mathbf{x})) = P(\mathbf{X}_{:;\mathbf{S}}) = \prod_{i \in I} P(\mathbf{X}_{i;\mathbf{S}}|\boldsymbol{PA}_{i;\mathbf{S}}^{\mathcal{G}}) \prod_{i \notin I} P(\mathbf{X}_{i;\mathbf{S}}|\boldsymbol{PA}_{i;\mathbf{S}}^{\mathcal{G}}) \tag{32}$$

*By do-operator can be used as normal conditional probability,*

$$P(\mathbf{X}_{\neg\mathbf{I};n}|do(\mathbf{X}_{\mathbf{I};n} = \hat{\mathbf{x}}), \mathbf{X}_{:;\mathbf{S}}) = \frac{P(\mathbf{X}_{:;\mathbf{S}\cup\{n\}}|do(\mathbf{X}_{I;n} = \hat{\mathbf{x}}))}{P(\mathbf{X}_{:;\mathbf{S}}|do(\mathbf{X}_{I;n} = \hat{\mathbf{x}}))} \tag{33}$$

$$= \frac{\prod_{i \in I} P(\mathbf{X}_{i;\mathbf{S}}|\boldsymbol{PA}_{i;\mathbf{S}}^{\mathcal{G}}) \prod_{i \notin I} P(\mathbf{X}_{i;\mathbf{S}\cup\{n\}}|\boldsymbol{PA}_{i;\mathbf{S}\cup\{n\}}^{\mathcal{G}})\big|_{\mathbf{X}_{I;n}=\hat{\mathbf{x}}}}{\prod_{i \in I} P(\mathbf{X}_{i;\mathbf{S}}|\boldsymbol{PA}_{i;\mathbf{S}}^{\mathcal{G}}) \prod_{i \notin I} P(\mathbf{X}_{i;\mathbf{S}}|\boldsymbol{PA}_{i;\mathbf{S}}^{\mathcal{G}})} \tag{34}$$

$$= \prod_{i \notin I} \frac{P(\mathbf{X}_{i;\mathbf{S}\cup\{n\}}|\boldsymbol{PA}_{i;\mathbf{S}\cup\{n\}}^{\mathcal{G}})\big|_{\mathbf{X}_{I;n}=\hat{\mathbf{x}}}}{P(\mathbf{X}_{i;\mathbf{S}}|\boldsymbol{PA}_{i;\mathbf{S}}^{\mathcal{G}})} \tag{35}$$

*By $P$ is the distribution for some ICM generative process, it satisfies the conditions of causal de Finetti theorem (multivariate) version:*

$$\mathbf{X}_{i;[n]} \perp\!\!\!\perp \mathbf{ND}^{\mathcal{G}}_{i;n+1} | \mathbf{PA}^{\mathcal{G}}_{i;[n]}$$

*where $\mathbf{PA}_i$ selects parents of node $i$ and $\mathbf{ND}_i$ selects non-descendants of node $i$ in $\mathcal{G}$. By exchangeability, one can use arbitrary set other than $[n]$, here use $\mathbf{S}$. Applying above conditional independence, we have*

$$\mathbf{X}_{i;\mathbf{S}} \perp\!\!\!\perp \mathbf{PA}^{\mathcal{G}}_{i;n} | \mathbf{PA}^{\mathcal{G}}_{i;\mathbf{S}}$$

*as $\mathbf{PA}^{\mathcal{G}}_{i;n} \subseteq \mathbf{ND}^{\mathcal{G}}_{i;n}$. Therefore,*

$$P(\mathbf{X}_{i;\mathbf{S}} | \mathbf{PA}^{\mathcal{G}}_{i;\mathbf{S}}) = P(\mathbf{X}_{i;\mathbf{S}} | \mathbf{PA}^{\mathcal{G}}_{i;\mathbf{S}\cup\{n\}}). \tag{36}$$

*With properties of conditional probability,*

$$\frac{P(\mathbf{X}_{i;\mathbf{S}\cup\{n\}} | \mathbf{PA}^{\mathcal{G}}_{i;\mathbf{S}\cup\{n\}})\big|_{\mathbf{X}_{I;n}=\hat{\mathbf{x}}}}{P(\mathbf{X}_{i;\mathbf{S}} | \mathbf{PA}^{\mathcal{G}}_{i;\mathbf{S}})} = P(X_{i;n} | \mathbf{X}_{i;\mathbf{S}}, \mathbf{PA}^{\mathcal{G}}_{i;\mathbf{S}\cup\{n\}})\big|_{\mathbf{X}_{I;n}=\hat{\mathbf{x}}} \tag{37}$$

*The result follows.*

## E    Conditioned interventional effect on other experiments

Under i.i.d. generative processes, if different interventions are placed on the system, then one often considers that data is coming from different environments and observing one experiment result cannot provide extra information about other experiments performed on random variables in different positions of the sequence. For example, consider a sequence of random variables generated by i.i.d., i.e., $(X_1, Y_1, Z_1), (X_2, Y_2, Z_2), \dots$ such that $X_n \to Y_n \to Z_n$ for all $n$. Then,

**i.i.d. generative processes** $: P(Y_1 | do(X_1 = x), do(Y_2 = y), Z_2) = P(Y_1 | do(X_1 = x)) \tag{38}$

due to $(X_1, Y_1, Z_1) \perp\!\!\!\perp (X_2, Y_2, Z_2)$. Therefore interventions on inconsistent intervention targets across the i.i.d. generated sequence can be equivalently considered as a mixture of data sampled from different environments, i.e., identical copies of SCMs with different hard-interventions.

However, this is not the case in ICM generative processes. Continuing the previous example, and let the sequence be data generated from an ICM generative process. Then,

**ICM generative processes** $: P(Y_1 = y_1 | do(X_1 = x), do(Y_2 = y), Z_2 = z_2)$

$$= \int p(y_1 | x, \theta_Z) p(\theta_Z | z_2, y) d\theta_Z \tag{39}$$

Eq. 39 explicitly shows that knowing the intervention effect on $Y_2$ acting on $Z_2$ helps one to infer the causal de Finetti parameter $\theta_Y$ and thus the intervention effect of $X_1$ on $Y_1$.

### E.1    Derivation of Eq. 39

Consider a DAG $\mathcal{G} := X \to Y \to Z$ and a distribution $P$ generated by the ICM generative process corresponding to $ICM(\mathcal{G})$. By truncated factorization in Theorem 1, we have:

$$P(Z_1 = z_1, Y_1 = y_1, X_1 = x, Z_2 = z_2, Y_2 = y, X_2 = x_2 | do(X_1 = \hat{x}), do(Y_2 = \hat{y})) \tag{40}$$

$$= \underbrace{\int \delta(X_1 = \hat{x}) p(x_2 | \theta_X) p(\theta_X) d\theta_X}_{:=p(x_2)} \underbrace{\int p(y_1 | \hat{x}, \theta_Y) \delta(Y_2 = \hat{y}) p(\theta_Y) d\theta_Y}_{:=p(y_1 | \hat{x})} \int p(z_2 | \hat{y}, \theta_Z) p(z_1 | y_1, \theta_Z) d\theta_Z$$

$$\tag{41}$$

$$= p(x_2) \int p(z_2 | \hat{y}, \theta_Z) p(z_1 | y_1, \theta_Z) p(y_1 | \hat{x}) p(\theta_Z) d\theta_Z \tag{42}$$

$$= p(x_2) \int p(z_2, \theta_Z | \hat{y}) p(z_1, y_1 | \hat{x}, \theta_Z) d\theta_Z \tag{43}$$

The merge of $p(z_2 | \hat{y}, \theta_Z) p(\theta_Z) = p(z_2, \theta_Z | \hat{y})$ is due to $\theta_Z$ is independent of $Y_n$ for any $n$ as $Z_n$ is always a collider in the path. The merge of $p(z_1 | y_1, \theta_Z) p(y_1 | \hat{x}) = p(z_1, y_1 | \hat{x}, \theta_Z)$ is due to $Z_1 \perp\!\!\!\perp X_1 | \theta_Z, Y_1$. Marginalizing away $X_2, Z_1$ and note $P(Z_2 = z_2 | do(X_1 = x), do(Y_2 = y)) = P(Z_2 = z_2 | do(X_1 = x))$ by Lemma 2. We have the desired result.

## F   Causal Pólya Urn Model

### F.1   Exchangeability of causal Pólya urn model

We first want to show exchangeability of the pair of the random variables $(X_i, Y_i)$ in the sequence $X_1, Y_1, X_2, Y_2, \ldots, X_n, Y_n$ that are generated via a causal Pólya urn model. Assume $n$ pairs of $(X_i, Y_i)$ are observed: out of $n$ balls we observe $n_1$ times that $X_n = 1, Y_n = 1$, $n_2$ times that $X_n = 1, Y_n = 0$, $n_3$ times that $X_n = 0, Y_n = 1$ and $n_4$ times that $X_n = 0, Y_n = 0$ where $n_1 + n_2 + n_3 + n_4 = n$. Note for the first time the number of balls in the urn for left and right compartment is $\alpha + \beta$ each, for the second draw $\alpha + \beta + 1$ each, and for the $k$-th draw $\alpha + \beta + k - 1$ each. The probability that we draw $(X_n = 1, Y_n = 1)$ pairs first, followed by $(X_n = 1, Y_n = 0)$ second, and then $(X_n = 0, Y_n = 1)$ third, followed by $(X_n = 0, Y_n = 0)$ last is:

$$P(X_1 = 1, Y_1 = 1, \ldots, X_n = 0, Y_n = 0) \tag{44}$$

$$= (\frac{\alpha}{\alpha + \beta} \times \frac{\beta}{\alpha + \beta}) \times (\frac{\alpha + 1}{\alpha + \beta + 1} \times \frac{\beta + 1}{\alpha + \beta + 1}) \times \cdots \times (\frac{\alpha + n_1 - 1}{\alpha + \beta + n_1 - 1} \times \frac{\beta + n_1 - 1}{\alpha + \beta + n_1 - 1}) \tag{45}$$

$$\times (\frac{\alpha + n_1}{\alpha + \beta + n_1} \times \frac{\alpha}{\alpha + \beta + n_1}) \times \cdots \times (\frac{\alpha + n_1 + n_2 - 1}{\alpha + \beta + n_1 + n_2 - 1} \times \frac{\alpha + n_2 - 1}{\alpha + \beta + n_1 + n_2 - 1}) \tag{46}$$

$$\times (\frac{\beta}{\alpha + \beta + n_1 + n_2} \times \frac{\alpha + n_2}{\alpha + \beta + n_1 + n_2}) \times \cdots \times (\frac{\beta + n_3 - 1}{\alpha + \beta + n_1 + n_2 + n_3 - 1} \times \frac{\alpha + n_2 + n_3 - 1}{\alpha + \beta + n_1 + n_2 + n_3 - 1}) \tag{47}$$

$$\times (\frac{\beta + n_3}{\alpha + \beta + n_1 + n_2 + n_3} \times \frac{\beta + n_1}{\alpha + \beta + n_1 + n_2 + n_3}) \times \cdots \times (\frac{\beta + n_3 + n_4 - 1}{\alpha + \beta + n - 1} \times \frac{\beta + n_1 + n_4 - 1}{\alpha + \beta + n - 1}) \tag{48}$$

For any permutation of the order $\pi : [n] \to [n]$, the denominator will not change as each step, we put an additional ball in each of the left and right compartments of the urn regardless of the color of the ball observed.

If we observe $j$-th black ball, i.e., $X_n = 1$ in the left compartment and $l$-th $Y_n | X_n$ in the right compartment such that $Z_n = 1$ at step $m$, then the numerator in the probability is $(\alpha + j - 1) \times (\alpha + l - 1)$ with the denominator as $(\alpha + \beta + m - 1) \times (\alpha + \beta + m - 1)$. Therefore, for any sequence $x_1, y_1, \ldots, x_n, y_n$ if $X_i = 1$ occurs $m_1$ times and $Z_i = 1 = (1 - X_i) * Y_i + (1 - Y_i) * X_i$ occurs $m_2$ times, then the final probability will always be equal to

$$P(X_1 = x_1, Y_1 = y_1, \ldots, X_n = x_n, Y_n = y_n) \tag{49}$$

$$= \frac{\prod_{i=1}^{m_1}(\alpha + i - 1) \prod_{i=1}^{n-m_1}(\beta + i - 1) \prod_{j=1}^{m_2}(\alpha + j - 1) \prod_{j=1}^{n-m_2}(\beta + j - 1)}{\prod_{k=1}^{n}(\alpha + \beta + k - 1) \times (\alpha + \beta + k - 1)} \tag{50}$$

$$= \frac{(\alpha + m_1 - 1)!(\beta + n - m_1 - 1)!(\alpha + \beta - 1)!}{(\alpha - 1)!(\beta - 1)!(\alpha + \beta + n - 1)!} \frac{(\alpha + m_2 - 1)!(\beta + n - m_2 - 1)!(\alpha + \beta - 1)!}{(\alpha - 1)!(\beta - 1)!(\alpha + \beta + n - 1)!} \tag{51}$$

### F.2   Conditional independences of causal Pólya urn model

For any $n \in \mathbb{N}$, we show $Y_{[n]} \perp\!\!\!\perp X_{n+1} \mid X_{[n]}$, i.e., $P(Y_{[n]} \mid X_{[n]}) = P(Y_{[n]} \mid X_{[n]}, X_{n+1})$.

By conditional probability is the division of two joints, we have

$$P(Y_n = y_n, \ldots, Y_1 = y_1 \mid X_n = x_n, \ldots, X_1 = x_1) = \frac{P(X_1 = x_1, Y_1 = y_1, \ldots, X_n = x_n, Y_n = y_n)}{P(X_1 = x_1, \ldots, X_n = x_n)} \tag{52}$$

$$= \frac{\prod_{i=1}^{m_1}(\alpha + i - 1) \prod_{i=1}^{n-m_1}(\beta + i - 1) \prod_{j=1}^{m_2}(\alpha + j - 1) \prod_{j=1}^{n-m_2}(\beta + j - 1)}{\prod_{k=1}^{n}(\alpha + \beta + k - 1) \times (\alpha + \beta + k - 1)} \bigg/ \frac{\prod_{i=1}^{m_1}(\alpha + i - 1) \prod_{i=1}^{n-m_1}(\beta + i - 1)}{\prod_{k=1}^{n}(\alpha + \beta + k - 1)} \tag{53}$$

$$= \frac{\prod_{i=1}^{m_2}(\alpha + i - 1) \prod_{i=1}^{n-m_2}(\beta + i - 1)}{\prod_{k=1}^{n}(\alpha + \beta + k - 1)} \tag{54}$$

To compute the joint probability of

$$P(X_1 = x_1, Y_1 = y_1, \ldots, X_n = x_n, Y_n = y_n, X_{n+1} = x_{n+1}) \tag{55}$$

$$= \frac{\prod_{i=1}^{m_1}(\alpha + i - 1)\prod_{i=1}^{n-m_1}(\beta + i - 1)\prod_{j=1}^{m_2}(\alpha + j - 1)\prod_{j=1}^{n-m_2}(\beta + j - 1)}{\prod_{k=1}^{n}(\alpha + \beta + k - 1) \times (\alpha + \beta + k - 1)} \tag{56}$$

$$\times \frac{(\alpha + m_1)\mathbb{1}_{x_{n+1}=1} + (\beta + n - m_1)\mathbb{1}_{x_{n+1}=0}}{\alpha + \beta + n} \tag{57}$$

Dividing $P(X_1 = x_1, \ldots, X_n = x_n, X_{n+1} = x_{n+1})$ leads to the same result.

## F.3   Application of causal de Finetti theorems

According to causal de Finetti theorems, given the causal Pólya urn model satisfies both exchangeability and certain conditional independence relationships, there must exist two unique independent prior functions that allow the joint distribution to be factored into two independent products. Here we provide an explicit representation. Let $B(\theta; \alpha, \beta)$ denotes beta distribution with parameter $\alpha, \beta$ and $B(\theta; \alpha, \beta) = \frac{(\alpha+\beta-1)!}{(\alpha-1)!(\beta-1)!}\theta^{\alpha-1}(1-\theta)^{\beta-1}$

$$P(X_1 = x_1, Y_1 = y_1, \ldots, X_n = x_n, Y_n = y_n) \tag{58}$$

$$= \int \theta^{\sum_n x_i} \times (1-\theta)^{(n-\sum_n x_i)} B(\theta; \alpha, \beta) d\theta \times \tag{59}$$

$$\int \psi^{\sum_n [(1-x_n)*y_n + (1-y_n)*x_n]}(1-\psi)^{n-\sum_n[(1-x_n)*y_n+(1-y_n)*x_n]} B(\psi; \alpha, \beta) d\psi \tag{60}$$

$$= [\frac{(\alpha + \beta - 1)!}{(\alpha - 1)!(\beta - 1)!}]^2 \int \theta^{\alpha + \sum_n x_i - 1}(1-\theta)^{n - \sum_n x_i + \beta - 1} d\theta \tag{61}$$

$$\int \psi^{\alpha + \sum_n z_n - 1}(1-\psi)^{n - \sum_n z_n + \beta - 1} d\psi, \text{where } z_n = (1-x_n)*y_n + (1-y_n)*x_n \tag{62}$$

note $\sum_n x_n = m_1, \sum_n z_n = m_2,$ by conjugate priors with bernoulli and beta distributions $\tag{63}$

$$= [\frac{(\alpha + \beta - 1)!}{(\alpha - 1)!(\beta - 1)!}]^2 \frac{\Gamma(\beta + n - m_1)\Gamma(\alpha + m_1)}{\Gamma(\alpha + \beta + n)} \frac{\Gamma(\beta + n - m_2)\Gamma(\alpha + m_2)}{\Gamma(\alpha + \beta + n)} \tag{64}$$

$$= \frac{(\alpha + m_1 - 1)!(\beta + n - m_1 - 1)!(\alpha + \beta - 1)!}{(\alpha - 1)!(\beta - 1)!(\alpha + \beta + n - 1)!} \frac{(\alpha + m_2 - 1)!(\beta + n - m_2 - 1)!(\alpha + \beta - 1)!}{(\alpha - 1)!(\beta - 1)!(\alpha + \beta + n - 1)!} \tag{65}$$

# G   Lemma 2 and its proof

**Lemma 2 (Intervention effect on differing positions)** *Given a graph $\mathcal{G}$ and let $P$ be the distribution for the ICM generative process with respect to $\mathcal{G}$. Let $\mathbf{X}$ and $\mathbf{Y}$ be two disjoint sets such that $\mathbf{X}$ is the intervention set and $\mathbf{Y}$ is the target set. Let $\mathbf{N_X} := \{n : X_{i;n} \in \mathbf{X}\}$ be the set of position indices being intervened, and similarly $\mathbf{N_Y}$ be the set of position indices being targeted to observe. Assume $\mathbf{N_X} \cap \mathbf{N_Y} = \emptyset$. Then,*

$$P(\mathbf{Y} \mid do(\mathbf{X} = \mathbf{x})) = P(\mathbf{Y}), \tag{16}$$

**Proof of Lemma 2** *Given an ICM generative process, the sequence is exchangeable by definition. By causal de Finetti's theorems, the joint distribution can be represented as*

$$p(\mathbf{x}_{:;1}, \ldots, \mathbf{x}_{:;N}) = \int \ldots \int \prod_n p(\mathbf{x}_{:;n}|\boldsymbol{\theta}_1, \ldots, \boldsymbol{\theta}_d) d\nu(\boldsymbol{\theta}_1) \ldots d\nu(\boldsymbol{\theta}_d) \tag{66}$$

$$= \int \ldots \int \prod_{n \in \mathbf{N_Y}} p(\mathbf{x}_{:;n}|\boldsymbol{\theta}_1, \ldots, \boldsymbol{\theta}_d) \prod_{n \notin \mathbf{N_Y}} p(\mathbf{x}_{:;n}|\boldsymbol{\theta}_1, \ldots, \boldsymbol{\theta}_d) d\nu(\boldsymbol{\theta}_1) \ldots d\nu(\boldsymbol{\theta}_d) \tag{67}$$

*The post-interventional distribution is:*

$$p(\mathbf{x}_{:;1}, \ldots, \mathbf{x}_{:;N} | do(\mathbf{X} = x)) \tag{68}$$

$$= \int \ldots \int \prod_{n \in \mathbf{N_Y}} p(\mathbf{x}_{:;n} | do(\mathbf{X} = \mathbf{x}), \boldsymbol{\theta}_1, \ldots, \boldsymbol{\theta}_d) \prod_{n \notin \mathbf{N_Y}} p(\mathbf{x}_{:;n} | do(\mathbf{X} = \mathbf{x}), \boldsymbol{\theta}_1, \ldots, \boldsymbol{\theta}_d) d\nu(\boldsymbol{\theta}_1) \ldots d\nu(\boldsymbol{\theta}_d) \tag{69}$$

$$= \int \ldots \int \prod_{n \in \mathbf{N_Y}} p(\mathbf{x}_{:;n} | \boldsymbol{\theta}_1, \ldots, \boldsymbol{\theta}_d) \prod_{n \notin \mathbf{N_Y}} p(\mathbf{x}_{:;n} | do(\mathbf{X} = \mathbf{x}), \boldsymbol{\theta}_1, \ldots, \boldsymbol{\theta}_d) d\nu(\boldsymbol{\theta}_1) \ldots d\nu(\boldsymbol{\theta}_d) \tag{70}$$

*The second equality follows from Definition 3 on the operational meaning of do-operator in ICM generative processes: since $\mathbf{N_X} \cap \mathbf{N_Y}$, there is no overlapping variables between $\mathbf{X}_{:;n}$ and $\mathbf{X}$, for all $n \in \mathbf{N_Y}$, which leads to no change of probability distribution.*

*Next, we proceed to integrating out every variable in $\mathbf{X}_{:;n}$ for $n \notin \mathbf{N_Y}$. Due to do-operation replaces intervened variable's conditional distribution as delta-distributions and keep the rest as conditional distributions while enforcing consistency over the intervened set of variables, the post-interventional distribution is in fact a conditional distribution by Theorem 1 with enforced consistency. Thus integration on all variables lead to summation 1.*

$$\int \ldots \int \prod_{n \notin \mathbf{N_Y}} p(\mathbf{x}_{:;n} | do(\mathbf{X} = \mathbf{x}), \boldsymbol{\theta}_1, \ldots, \boldsymbol{\theta}_d) d\mathbf{x}_{:;n} = 1 \tag{71}$$

*Therefore,*

$$p(\mathbf{x}_{:;\mathbf{N_Y}} | do(\mathbf{X} = \mathbf{x})) = \int \ldots \int \prod_{n \in \mathbf{N_Y}} p(\mathbf{x}_{:;n} | \boldsymbol{\theta}_1, \ldots, \boldsymbol{\theta}_d) d\nu(\boldsymbol{\theta}_1) \ldots d\nu(\boldsymbol{\theta}_d) \tag{72}$$

$$= p(\mathbf{x}_{:;\mathbf{N_Y}}) \tag{73}$$

*Marginalizing out every other variable in $\mathbf{X}_{:;\mathbf{N_Y}}$ except the target variables in $\mathbf{Y}$, the result follows.*

## H   Lemma 3 and its proof

**Lemma 3 (Intervention effect within the same position)** *Given a graph $\mathcal{G}$ and let $P$ be the distribution for the ICM generative process with respect to $\mathcal{G}$. Let $\mathbf{X}$ be the intervention set such that it consists only $\mathbf{X}_{\mathbf{I};n}$ where $n \in \mathbf{S} \subseteq [N]$ and $\mathbf{I} \subseteq [d]$ is a set of variable indices. Let $\mathbf{Y}$ be the target set such that it consists only $\mathbf{X}_{\mathbf{J};n}$ where $n \in \mathbf{S} \subseteq [N]$ and $\mathbf{J} \subseteq [d]$. Note $\mathbf{I} \cap \mathbf{J} = \emptyset$. Then,*

$$P(\mathbf{Y} \mid do(\mathbf{X} = \hat{\mathbf{x}})) = \sum_{\boldsymbol{pa}_{\mathbf{X}}} P(\mathbf{Y} \mid \hat{\mathbf{x}}, \boldsymbol{pa}_{\mathbf{X}}) P(\boldsymbol{pa}_{\mathbf{X}}), \tag{17}$$

*where $\boldsymbol{PA}_{\mathbf{X}}$ denotes the parent set of intervened variables $\mathbf{X}$.*

**Proof of Lemma 3** *The proof follows a similar idea as in the case of i.i.d. data, here we include it for completeness. Note the joint distribution can be written as (by argument in Proof of Theorem 1)*

$$P(\mathbf{X}_{:;\mathbf{N}}) = \prod_i P(\mathbf{X}_{i;\mathbf{N}} | \boldsymbol{PA}_{i;\mathbf{N}}). \tag{74}$$

*Then when intervening on $\mathbf{X}$ which shares the identical set of variable indices across different samples, do-operation means assigning $P(\mathbf{X}_{i;\mathbf{N}} | \boldsymbol{PA}_{i;\mathbf{N}})$ to delta-distribution for all $i \in \mathbf{I}$. This can equivalently be expressed as a division of conditional distributions:*

$$P(\mathbf{X}_{:;\mathbf{N}} | do(\mathbf{X} = \mathbf{x})) = \begin{cases} \frac{P(\mathbf{X}_{:;\mathbf{N}})}{\prod_{i \in \mathbf{I}} P(\mathbf{x}_{i;\mathbf{N}} | \boldsymbol{PA}_{i;\mathbf{N}})} & \text{when } \mathbf{X} = \mathbf{x} \\ 0 & \text{else} \end{cases} \tag{75}$$

*Next we proceed to show*

$$\prod_{i \in \mathbf{I}} P(\mathbf{X}_{i;\mathbf{N}} | \boldsymbol{PA}_{i;\mathbf{N}}) = P(\mathbf{X}_{\mathbf{I};\mathbf{N}} | \boldsymbol{PA}_{\mathbf{I};\mathbf{N}}) = P(\mathbf{X} | \boldsymbol{PA}_{\mathbf{X}})$$

*Without loss of generality, we can always order $\mathbf{X_{I;N}}$ according to reversed topological ordering such that all $X_i$'s non-descendants will be placed after the variable index $i$. Note if $X_j$ is the non-descendant of $X_i$, then so is $\mathbf{PA}_j$. If not, suppose some variable in $\mathbf{PA}_j$ is a descendant of $X_i$, then $X_j$ is a descendant of that variable, so $X_j$ is a descendant of $X_i$, contradiction. This means after re-ordering due to $X_{i;\mathbf{N}} \perp\!\!\!\perp ND_{i;\mathbf{N}}|\mathbf{PA}_{i;\mathbf{N}}$, where $ND_{i;\mathbf{N}} = \mathbf{PA}_{\mathbf{I}\setminus\{i\};\mathbf{N}} \cup \mathbf{X}_{>i;\mathbf{N}}$ in this case, we have:*

$$P(\mathbf{X_{I;N}}|\mathbf{PA_{I;N}}) = \prod_i P(\mathbf{X}_{i;\mathbf{N}}|\mathbf{PA_{I;N}}, \mathbf{X}_{>i;\mathbf{N}}) = \prod_{i \in \mathbf{I}} P(\mathbf{X}_{i;\mathbf{N}}|\mathbf{PA}_{i;\mathbf{N}}) \tag{76}$$

*Then,*

$$P(\mathbf{X}_{:;\mathbf{N}}|do(\mathbf{X} = \hat{\mathbf{x}})) = \frac{P(\mathbf{X}_{:;\mathbf{N}})}{P(\hat{\mathbf{x}}, \mathbf{PA_X})} * P(\mathbf{PA_X}) \tag{77}$$

*Integrating on every variable other than those contained in $\mathbf{Y}$, we have*

$$P(\mathbf{Y}|do(\mathbf{X} = \hat{\mathbf{x}})) = \sum_{\mathbf{pa_X}} P(\mathbf{Y}|\hat{\mathbf{x}}, \mathbf{pa_X})P(\mathbf{pa_X}) \tag{78}$$

## I *Do-Finetti* Algorithm

Do Finetti algorithm recovers graphical estimation and causal effect estimation simultaneously through collecting multi-environment data, that is marginal copies of an exchangeable process. The algorithm combines *Causal-de-Finetti* algorithm developed in Guo et al. [2023a] and the truncated factorization for ICM generative processes developed in Section 3 of this paper.

---

**Algorithm 1** "Do-Finetti" Algorithm: causal effect estimation in ICM-generative processes

---

**Input:** For $e \in \mathcal{E}$, we have $(X_{1;n}^e, \ldots, X_{d;n}^e)_{n=1}^N$ where $X_{i;n}^e$ denotes the $i$-th variable observed at $n$-th position in environment $e$. Let $N$ denote the number of observations in each environment $e$. Assume $N \geq 2$. Let $\mathbf{X}$ and $\mathbf{Y}$ be two disjoint subsets of variables such that $\mathbf{X}, \mathbf{Y} \subseteq \{(X_{1;n}, \ldots, X_{d;n})\}_{n=1}^N$.
**Output:** $P(\mathbf{Y} = \mathbf{y}|do(\mathbf{X} = \mathbf{x}))$
**Step 1:** Run *Causal-de-Finetti* Algorithm on multi-environment data and return estimated $\hat{\mathcal{G}}$.
**Step 2:** Estimate the probability distribution from grouped data $\hat{P}(\mathbf{X}_{:;1}, \ldots, \mathbf{X}_{:;N})$, e.g., histograms for discrete variables or kernel density estimation [Simonoff, 1996] for continuous variables.
**Step 3:** Marginalize out variables not included in $\mathbf{X} \cup \mathbf{Y}$ according to the truncated factorization for ICM generative processes with $\mathcal{G} = \hat{\mathcal{G}}$ (see Eq. 12 or below):

$$p(\mathbf{x}_{:;1}, \ldots, \mathbf{x}_{:;N}|do(\mathbf{X} = \mathbf{x})) = \prod_{i \in I_{\mathbf{X}}} p(\mathbf{x}_{i;[\neg\mathbf{N}_i]}|\mathbf{pa}_{i;[\neg\mathbf{N}_i]}^{\mathcal{G}}) \prod_{i \notin I_{\mathbf{X}}} p(\mathbf{x}_{i;[N]}|\mathbf{pa}_{i;[N]}^{\mathcal{G}})\big|_{\mathbf{X} = \mathbf{x}}$$

**Step 4:** Return $P(\mathbf{Y} = \mathbf{y}|do(\mathbf{X} = \mathbf{x}))$

---

## J  Proof of Theorem 2

**Theorem 4 (Unique graphical identification theorem [Guo et al., 2023a])** *Consider the set of distributions that are both Markov and faithful to $ICM(\mathcal{G})$, denoted as $\mathcal{E}(\mathcal{G})$. Then,*

$$\mathcal{E}(\mathcal{G}_1) = \mathcal{E}(\mathcal{G}_2) \text{ if and only if } \mathcal{G}_1 = \mathcal{G}_2 \tag{79}$$

**Theorem 2 (Causal effect identification in ICM generative processes)** *Denote $\mathbf{Y}, \mathbf{X}$ be two disjoint subsets of observable variables in $\{\mathbf{X}_{:;n}\}_{n\in\mathbb{N}}$. Then $P(\mathbf{Y}|do(\mathbf{X} = \mathbf{x}))$ is identifiable given the distribution $P$ from the class of distributions generated from ICM generative processes. Here identifiability means the causal query can be computed uniquely from $P$.*

**Proof of Theorem 2** *Denote $\mathcal{P}_G$ be the set of distributions that is Markov and faithful to the DAG $G$. Consider a statistical model $\mathcal{P} := \{\mathcal{P}_G : G \text{ is a DAG}\}$. We say $G$ is identifiable from $\mathcal{P}$ if $G \to \mathcal{P}_G$ is one-to-one, i.e., $\mathcal{P}_{G_1} = \mathcal{P}_{G_2}$ if $G_1 = G_2$. This directly follows from Theorem 5 in [Guo et al., 2023a]. As $G$ is identifiable from $P$ theoretically, given the pair $P, G$, by Theorem 1 in this paper, we know the truncated factorization over all observable variables. Through marginalization for arbitrary target set $\mathbf{Y}$ and arbitrary intervention set $\mathbf{X}$, the causal query is uniquely computable.*

## K   Experimental Details

The data-generating process is described as below for each of the bivariate graph:

$$\theta^e \sim \text{Beta}(\alpha, \beta), \psi^e \sim \text{Beta}(\alpha, \beta)$$
$$X \rightarrow Y : X_i^e := \text{Ber}(\theta^e), Y_i^e := \text{Ber}(\psi^e) \oplus X_i^e$$
$$Y \rightarrow X : X_i^e := \text{Ber}(\theta^e) \oplus Y_i^e, Y_i^e := \text{Ber}(\psi^e)$$
$$X \perp\!\!\!\perp Y : X_i^e := \text{Ber}(\theta^e), Y_i^e := \text{Ber}(\psi^e)$$

where $\oplus$ denotes xor operation and $X_i^e, Y_i^e$ denotes variable generated at $i$-th position in environment $e$ and set $\alpha = 1, \beta = 3$.

The experiments can be reproduced using single laptop with CPUs with a time estimate within 5 minutes. The code is building on top of Guo et al. [2023a] under license CC-BY 4.0. For each experiment, we randomly generate data corresponding to one of the three graphs. We randomly choose the intervened variable $X = X_1, Y_1, X_2$ or $Y_2$ and randomly determine the intervened value $\hat{x}$ to be 0 or 1. Our goal is to estimate the post-interventional distribution $P(X_1, Y_1, X_2, Y_2 | do(X = \hat{x}))$ as accuracy as possible. Our mean squared error loss is the predicted $\hat{P}$ with analytic $P$ summing over all possible enumeration of $X_1, Y_1, X_2, Y_2$ values. We use PC algorithm for i.i.d. graph identification and truncated factorization as in Eq. 2 for effect estimatinon given estimated DAG. Similarly, we run *Do-Finetti* algorithm. We compare with the analytic solutions that can be derived due to Beta-bernoulli distributions are conjugate priors towards each other. Section K.1 shows an example of derivation. To clearly see errors in MSE loss in causal effect estimation is not purely driven by graphical misclassification, we provide true oracle DAG to each of the method – *Do-Finetti-w-true-dag* and *IID-w-true-dag* – and observe that even with inifinte data, IID can never achieves near-zero causal effect estimation errors as shown in Fig. 4a. This means the traditional truncated factorization fails to characterize causal effects in ICM generative processes.

### K.1   Analytic solution

We choose Beta and bernoulli distributions in particular due to they are conjugate prior properties to allow us to calculate easily their analytic solutions. Note the distributions can be arbitrary and we can approximate the integration via standard methods, e.g. histograms or kernel density estimations. Taking the example of $X \rightarrow Y$ generated under the data-generating process described in Experiment section, then by truncated factorization in ICM generative processes we have:

$$P(Y_1 | do(x_1), X_2, Y_2) = P(Y_1 | x_1, X_2, Y_2) \tag{80}$$

$$= \frac{P(Y_1, Y_2 | x_1, X_2)}{P(Y_2 | X_2)} \tag{81}$$

The second equality is due to ICM generative processes properties $Y_2 \perp\!\!\!\perp X_1 | X_2$. Note by causal de Finetti, we know:

$$P(y_1, y_2 | x_1, x_2) = \int p(y_i | x_i, \psi) p(\psi) d\psi \tag{82}$$

Given Bernoulli distribution, we can re-write the assignment function $Y_i^e = \text{Ber}(\psi^e) \oplus X_i^e$ in terms of probability distribution, where $z = (1 - x) * y + (1 - y) * x$ and $p(y | x, \psi) = \psi^z (1 - \psi)^{1-z}$. Note $p(\psi)$ follows a Beta distribution such that $p(\psi) = \frac{\psi^{\alpha-1}(1-\psi)^{\beta-1}}{B(\alpha,\beta)}$. The marginal likelihood from manipulation gives $P(y_1, y_2 | x_1, x_2) = \frac{B(\alpha_2, \beta_2)}{B(\alpha,\beta)}$, where $\alpha_N = \sum_n z_n + \alpha, \beta_N = N - \sum_n z_n + \beta$. Putting it all together,

$$P(y_1 | do(x_1), x_2, y_2) = \frac{B(\alpha + z_1 + z_2, 2 - z_1 - z_2 + \beta)}{B(\alpha + z_2, 1 + \beta - z_2)} \tag{83}$$

Recall $B(\alpha, \beta) = \frac{\Gamma(\alpha)\Gamma(\beta)}{\Gamma(\alpha+\beta)}$ and $\Gamma(n) = (n-1)!$. Rearranging gives:

$$P(y_1 = 0 | do(x_1 = 0), x_2 = 0, y_2 = 0) = \frac{B(\alpha, 2 + \beta)}{B(\alpha, 1 + \beta)} = \frac{\Gamma(\beta + 2)}{\Gamma(\alpha + \beta + 2)} \times \frac{\Gamma(\alpha + \beta + 1)}{\Gamma(\beta + 1)} \tag{84}$$

$$= \frac{\beta + 1}{\alpha + \beta + 1} = \frac{4}{5} = 0.8 \tag{85}$$

Similarly, $P(y_1|do(x_1)) = P(y_1|x_1) = B(\alpha + z_1, 1 + \beta - z_1)$. Then, $P(y_1 = 0|do(x_1 = 0)) = \frac{\Gamma(\alpha)\Gamma(1+\beta)}{\Gamma(1+\alpha+\beta)} = \frac{1}{4}$ when $\alpha = 1, \beta = 3$.

## L  Limitations and Broader Impacts

**Limitations** This work has taken first steps in formulating what intervention means in exchangeable data satisfying the ICM principle. However, much is left to do: from understanding interventions to counterfactual queries, from Markovian models to semi-Markovian models. We note there is a whole world of possibilities in formalizing causality in exchangeable data settings: both from the traditional causality view – e.g., [Karlsson and Krijthe, 2024] perform confounder detection in multi-environment data – and from the view of representation learning where [Reizinger et al., 2022] show that nonlinear ICA under certain assumptions can perform causal discovery in a de Finetti framework. We hope this work opens up possibilities to connect to interventional representation learning and the branches of causality research.

**Broader Impacts** This paper is a foundational research that studies causal interventions under exchangeable contexts. It is intended to advance the field of machine learning and through studying causality, this work intends to provide control and understanding over the world and the current increasingly more opaque and 'black-box' models to safeguard potential harms for society. However, we note, every innovation has the potential to be misused with malicious intent. For example, we foresee that once offering more control and understanding, users have the potential to have more precise manipulation. This paper at its current status still presents a huge gap between theory and practice.

