# OpenReview forum: "Do Finetti: On Causal Effects for Exchangeable Data"
_NeurIPS.cc/2024/Conference — NeurIPS 2024 oral_

### Official Review · Reviewer_nbBj · 2024-07-13

**Soundness:** 4
**Presentation:** 4
**Contribution:** 2
**Rating:** 7
**Confidence:** 2

**Summary:**

The paper generalizes the traditional iid settings in casual inference to exchangeability settings by de Finetti theorem, and proposes a new model named the casual Polya urn model to illustrate the new scheme and to catch more relationship. The experiments show when the number of environment is less than 5000, the new schemes performs well.

**Strengths:**

First of all, I am sorry that I do not know much about the casual inference. But the paper uses exchangeability instead of iid settings, which seems an improvement.

**Weaknesses:**

1. Aldous 1985 shows many (not all) conclusions in iid can be naturally transformed into those under exchangeability. So the theoretical improvement seems not much.
2. Only simulated experiments for the casual inference problems.
3. De Finetti Theorem is ‘iff’. So it is inappropriate to use methods based on iid settings on the exchangeable but not iid data to compare.
4. For the experiment, what about the larger number of environment? It seems the original one performs better.

**Questions:**

Besides above, 5. In casual inference problems, is it easy to identify the exchangeability, especially for the real data?

**Limitations:**

Besides above, 6. Some typos even in reference; for example, the first reference is not well-written. 7. Could see more complicated structure between theta psi and X. In the paper, psi is independent of X. 8. Typos, like ‘Nature’ should be ‘nature’.

---

> ### Author Rebuttal · Authors · 2024-08-05
>
> We thank the reviewer for taking the time and address the questions below.
> > Aldous 1985 shows many (not all) conclusions in iid can be naturally transformed into those under exchangeability. So the theoretical improvement seems not much.
>
> Indeed, many i.i.d. results transfer to the exchangeable case, however, the situation is more complex. Aldous (1985) presents a lecture on exchangeability in probability theory, covering topics such as de Finetti’s theorem, its consequences, extensions, analogues, distributions invariant under group transformations, and exchangeable sets. However, Aldous (1985) does not focus on causality. Our work shows that causal effect estimation in certain types of non-IID data (specifically those meeting an ICM assumption) enables us to draw conclusions that cannot be inferred from IID data alone.
>
> > Only simulated experiments for the casual inference problems.
>
> The experiments are designed to show that the truncated factorization formula developed for i.i.d. data fails to apply for exchangeable non-i.i.d. data even given knowledge of the true graph. Fig. 4 shows even with near infinite data, the dotted blue line (i.i.d. with the true graph) cannot reach 0 mean squared error in contrast to do-finetti with the true graph. The simulated experiment thus merely creates a controlled setup to demonstrate that conclusions in i.i.d. (e.g. truncated formula) fail to apply for exchangeable non-i.i.d. data.
>
> > De Finetti Theorem is ‘iff’. So it is inappropriate to use methods based on iid settings on the exchangeable but not iid data to compare.
>
> We agree. The standard i.i.d. methods aren’t appropriate. Maybe we have not expressed this well, and a statement akin to the above would help put the experiment in perspective?
>
> > For the experiment, what about the larger number of environment? It seems the original one performs better.
>
> We are confused with the statement. In Figure 4 (both left and right), we observe that do-Finetti algorithm outperforms the i.i.d. baseline in the large number of environments. The left plots show that do Finetti achieves near zero mean squared error in causal effect estimation compared to the i.i.d. baseline which has high errors. The right plot also shows that do Finetti simultaneously identifies the correct graph in contrast to the i.i.d. baseline with low graph accuracy.
>
> > 5. In causal inference problems, is it easy to identify the exchangeability, especially for the real data?
>
> We thank the reviewer for the question. There might be empirical tests one could run based on permutations and there exists some work for testing exchangeability in real-world data, e.g., [1], [2]. Likely, this may not be a closed question, especially in terms of causal inference and exchangeability. Though we’d hope to leverage the above work as a promising first step to study it for future directions.
>
> > 6. Some typos even in reference; for example, the first reference is not well-written.
>
> We apologise and will thoroughly go over this for the revision.
>
> > 7. Could see more complicated structure between theta psi and X. In the paper, psi is independent of X.
>
> This could be interesting future directions, however we see this as a starting point, and we found ICM assumption was ideal for us in that
> It allows us to derive non-trivial results (Theorem 2)
> It is an assumption which is common in the causality community [3].
>
> > 8. Typos, like ‘Nature’ should be ‘nature’.
>
> Here we deliberately choose the capital letter “Nature” to show respect and reverence towards the governing laws of the universe. This though common in the literature, we acknowledge it is more a personal preference.
>
> Overall, we thank the reviewer for taking the time and if we adequately addressed your concerns over
> * theoretical improvement (exchangeable non-i.i.d. data reveals important properties current causal literature does not cover), and
> * fair experimental comparisons and results (i.i.d. methods fail to apply for exchangeable non-i.i.d. data and hence demands a new causal effect estimation function for exchangeable non-i.i.d. data and do Finetti algorithm offers a solution),
>
> we invite the reviewer to consider raising the score.
>
>
> [1] Vovk, V., Gammerman, A., Shafer, G. (2022). Testing Exchangeability. In: Algorithmic Learning in a Random World. Springer, Cham. https://doi.org/10.1007/978-3-031-06649-8_8
>
> [2] Aw, Alan J., Jeffrey P. Spence, and Yun S. Song. "A simple and flexible test of sample exchangeability with applications to statistical genomics." The annals of applied statistics 18.1 (2024): 858.
>
> [3] Schölkopf, Bernhard, et al. "Toward causal representation learning." Proceedings of the IEEE 109.5 (2021): 612-634.

---

> > ### Comment · Reviewer_nbBj · 2024-08-11
> >
> > Thank you for your reply! Since I am not an expert of casual inference, my questions focus on the exchangeability. Since I don't find many literature about exchangeability on causal inference structure, combining your rebuttal, I agree that a non-iid structure is non-trivial and an interesting topic. Since in the exchangeability settings, more relationship with Bayesian methods could be explored more.

---

> > > ### Author Response · Authors · 2024-08-11
> > >
> > > We thank the reviewer for responding and pushing us to be clear on the paper's contributions! We agree with the reviewer that there could be exciting areas to explore on the connection between causality and Bayesian methods.

---

### Official Review · Reviewer_H65Y · 2024-07-13

**Soundness:** 4
**Presentation:** 3
**Contribution:** 3
**Rating:** 7
**Confidence:** 2

**Summary:**

The paper studies causal effect identification and estimation in exchangeable data. The main result here is theorem 1, which shows that causal effects are identifiable in ICM generative processes.

**Strengths:**

- The paper provides a great framework to think about interventions in exchangeable data. Starting from what interventions should be considered (Definition 3) to identifying a procedure for computing the post-interventional distributions.
- The paper presentation, at least in the first part, was simple and intuitive. I always found myself asking a question and then find it being answered in the next paragraph. However, probably due to space constraints, this did change in the latter parts of the paper.

**Weaknesses:**

- The latter parts of the paper is rushed and left me confused. For example, it is unclear  how causal de Finetti theorems apply to the Causal Pólya Urn Model, Theorem 2, and the entirety of section 4 is very rushed and I have struggled to understand what theorem 2 say exactly.
- I have felt that the algorithm could have taken more of real-estate in the presentation of the paper. Also, it is unclear how the graph structure is learned in the algorithm
- This is more of a nit pick, but the appendix contains a few typos and is in a worse state in general than the main text. For example, the use of index i in equation 51, 53, ...

**Questions:**

- In the experiments, it seems to me that the model generating the synthetic dataset is different from that described in section 3.2. In particular, in the experiments, X_i is sampled from a Ber(theta) and hence P(X_i = 1) = P(X_2 =1) = ... = theta, whereas if I understood the model in 3.2, then the probability P(X_n =1) will be much greater than P(X_1 =1) if for example all X_m =1 for all m < n. Are they actually different? Or did I misunderstood? And how can the model described in 3.2 be represented by equation 4? (I read F.2 but it seems to me that equation 51 follows the model in Section 5).

- In the experiments, can the authors elaborate on the IID baseline? Do you run the algorithm on the "full" graph G which have nodes X_1 Y_1 X_2 Y_2? I assume this is what's been done as it is the fairest baseline, but I'm not sure. Appendix K seems to imply that and the main paper did not make it clear.

- In the description of ICMs, the author mention the expression:
> Causal mechanisms are independent of each other in the sense that a change in one mechanism P(Xi | PAi) does not inform or influence any of the other mechanisms P(Xj | PAj)
What would be a concrete example where such condition is violated?

**Limitations:**

It has been addressed.

---

> ### Author Rebuttal · Authors · 2024-08-05
>
> We thank the reviewer for their time and their appreciation of our work. We hope to clarify their questions below:
>
> > The latter parts of the paper is rushed and left me confused. For example, it is unclear how causal de Finetti theorems apply to the Causal Pólya Urn Model, Theorem 2, and the entirety of section 4 is very rushed and I have struggled to understand what theorem 2 say exactly.
>
> We apologise and will try to clarify. Appendix F shows the causal Pólya urn model can be equivalently modelled as in the causal de Finetti theorem, i.e., $\int \int \prod_i p(y_i | x_i, \psi) p(x_i | \theta) p(\theta) p(\psi) d\theta d\psi$, where $p(\theta), p(\psi)$ are beta-distributions and $p(x_i | \theta), p(y_i | x_i, \psi)$ are Bernoulli distributions. This is a bivariate version of equation 4 when we only consider two variables X and Y. We will include a more detailed discussion on Appendix F in the main text for the next version.
>
> Theorem 2 says that for ICM generative processes, both causal graphs and causal effects can be identified simultaneously. This is in contrast to an i.i.d. process, where causal effect identification often requires access to aspects of the causal graph which itself is not identifiable from observational data, and thus it is assumed that the causal graph is provided in addition to the observational data.
>
> > Also, it is unclear how the graph structure is learned in the algorithm
>
> For learning the graph structure, we refer to Algorithm 1 in Guo et al. 2024 [1], as the present paper focuses on the study of causal effects. We will make this point clearer in the next version.
>
> > This is more of a nit pick, but the appendix contains a few typos and is in a worse state in general than the main text. For example, the use of index i in equation 51, 53, ...
>
> Thank you for pointing out the typos, we will correct them in next version.
>
> > In the experiments, it seems to me that the model generating the synthetic dataset is different from that described in section 3.2. In particular, in the experiments, X_i is sampled from a Ber(theta) and hence P(X_i = 1) = P(X_2 =1) = ... = theta, whereas if I understood the model in 3.2, then the probability P(X_n =1) will be much greater than P(X_1 =1) if for example all X_m =1 for all m < n. Are they actually different? Or did I misunderstood? And how can the model described in 3.2 be represented by equation 4? (I read F.2 but it seems to me that equation 51 follows the model in Section 5).
>
> The model described in 3.2 can be represented by equation 4, because Appendix F.2 shows that the joint distribution $P(x1, y1, x2, y2, …) $ in the causal Pólya urn model can be modelled as  $\int \int \prod_i p(y_i | x_i, \psi) p(x_i | \theta) p(\theta) p(\psi) d\theta d\psi$, where $p(\theta), p(\psi)$ are beta-distributions and $p(x_i | \theta), p(y_i | x_i, \psi)$ are Bernoulli distributions. This corresponds to equation 4 as this is the equivalent bivariate version where X is the parent of Y, and $\theta, \psi$ are statistically independent $\theta_i$’s in equation 4. Therefore as equation 51 follows the model in section 5 (as the reviewer suggested) and it is the representation for the causal Pólya urn model (due to the arguments above and in Appendix F.2), we argue that it is the same as the one described in section 3.2. We hope it clarifies things and thank you for going into the Appendix.
>
> > In the experiments, can the authors elaborate on the IID baseline? Do you run the algorithm on the "full" graph G which have nodes X_1 Y_1 X_2 Y_2?
>
> The IID baseline is taking into account the full graph x1, x2, y1, y2 and treats the variables as $(x_i, y_i) \sim_{i.i.d.} (X, Y)$. This means there are no bi-directed edges connecting X1, X2 and Y1, Y2. The causal effect estimand for the i.i.d. case is analogous to Eq. 10. The experiment is designed to show that the truncated factorization developed for i.i.d. data indeed does not apply for exchangeable non-i.i.d. data, hence the need for the generalised truncated factorization introduced in Theorem 1.
>
> > In the description of ICMs, the author mention the expression:
> Causal mechanisms are independent of each other in the sense that a change in one mechanism P(Xi | PAi) does not inform or influence any of the other mechanisms P(Xj | PAj) What would be a concrete example where such condition is violated?
>
> This condition will be violated when we decompose a distribution into non-causal conditionals. For example, suppose that for weather stations, altitude (A) causes temperature (T) but not vice versa, i.e. building a greenhouse effect on top of a mountain will not increase the height of the mountain. In that case, P(T | A) and P(A) will be independent causal mechanisms that can be changed independently in the generative process, but P( A | T) and P(T) will not be. This is described in more detail for instance in Peters et al., Elements of Causal Inference. The causal de Finetti theorem formalises ICM mathematically: suppose $\theta$ represents a mountain and $\psi$ represents seasons. With random measurements given fixed mountain and season, we have $T_i, A_i$. The causal de Finetti theorem says $A \to T$ characterizes by $T_1 \perp A_2 | A_1$. Equivalently, it means $P(T_1 | A_1, A_2) = P(T_1 | A_1)$, i.e. knowing the altitude of measurements in other locations will not help the prediction of the temperature measured at location 1. If $T \to A$, causal de Finetti theorem says $A_1 \perp T_2 | T_1$, equivalently expressed as $P(A_1 | T_1, T_2) = P(A_1 | T_1)$. However we know it does not hold as if $T_1 = -10$, and $T_2 = 30$, then one can infer it is likely to be in hot season and given observed a low temperature $T_1$, one could infer $A_1$ will be high in altitude.
>
> [1] Guo, S., Tóth, V., Schölkopf, B. and Huszár, F., 2024. Causal de Finetti: On the identification of invariant causal structure in exchangeable data. Advances in Neural Information Processing Systems, 36.

---

> ### Comment · Reviewer_H65Y · 2024-08-10
>
> I thank the reviewer for their comprehensive and very well-explained response. No further questions from me!

---

> > ### Author Response · Authors · 2024-08-11
> >
> > Thank you for taking the time and helping us to improve the paper! We will include the clarification on causal Pólya urn model in the main text for the next revision.

---

### Official Review · Reviewer_Jkyk · 2024-07-30

**Soundness:** 3
**Presentation:** 4
**Contribution:** 4
**Rating:** 8
**Confidence:** 4

**Summary:**

The paper formalizes the observational and interventional distribution under the ICM generative process, of which iid is the special case. It provides an identifiability result for the causal effect given that the causal graph is known. Then, it shows that both the causal graph and the causal effect can be identified simultaneously.

**Strengths:**

1. Problem: The problem is important as it will bring the causal effect estimation literature closer to real-world scenarios.

2. Theory: The theoretical results are strong, especially Theorem 2, which shows that both the causal graph and the effect can be estimated simultaneously. I have not checked the proofs, though.

3. Experiment: The experiment on the simulated data verify the theoretical claim.

4. Presentation: The paper is well-written and easy to follow. All the notation and definitions are clear.

**Weaknesses:**

1. Experiments: I understand the main purpose of the work is to establish the theoretical foundation of causal effect estimation for exchangeable data, but it would be interesting to apply the method to some real-world datasets (not necessary for the rebuttal).

**Questions:**

1. Definition 3: We should also break the edge from the de-finnetti parameters to the intervened variable, right? Or do we not need any graphical operations?

2. I am slightly confused by the statements in Lines 153-154 and Line 197. They seem to contradict each other. Is it due to conditioning on x1  and x2 that Eq10 and eq11 are not equal since, in ICM, they are not iid?

**Limitations:**

Yes, the authors have addressed the limitation in the Conclusion section and Appendix L.

---

> ### Author Rebuttal · Authors · 2024-08-05
>
> We thank the reviewer for their time and their recognition of the importance of relaxing the i.i.d. assumption, a general problem in the ML community. Please see below answers to the  questions:
>
> > Definition 3: We should also break the edge from the de-finnetti parameters to the intervened variable, right? Or do we not need any graphical operations?
>
> Yes, we need to break the edge from the de-Finetti parameters to the intervened variables when performing an intervention. This definition aims to clarify different implications when performing graph surgery on SCM and ICM processes.
>
> > I am slightly confused by the statements in Lines 153-154 and Line 197. They seem to contradict each other. Is it due to conditioning on x1 and x2 that Eq10 and eq11 are not equal since, in ICM, they are not iid?
>
> Lines 153-154 state that IID is a special case of exchangeability whenever p(ψ) = δ(ψ = ψ0), and line 197 states that the causal effect differs whenever p(ψ) does not equal δ(ψ = ψ0). These statements are consistent in that IID is a special case of ICM when $ p(\psi) = \delta(\psi = \psi_0) $. However, our focus is on causal effects for the ICM exchangeable non-IID case, where $p(\psi) \neq \delta(\psi = \psi_0)$. We show that the causal effects in the ICM exchangeable non-IID case (Eq. 11) differ from those in the ICM IID case (Eq. 10) due to the dependency among observations.

---

> > ### Comment · Reviewer_Jkyk · 2024-08-12
> >
> > I thank the authors for their response. I will keep the score.

---

### Decision · Program_Chairs · 2024-09-25

**Decision:**

Accept (oral)

**Comment:**

Reviewers were agreed in their support for accepting this paper, praising the topic, the strength of the results, and the clarity of presentation (particularly in the early parts of the paper).  The authors should revise the later parts of the paper, consistent with the comments of reviewer H65Y, including providing more detail on how the causal de Finetti theorems apply to the Causal Pólya Urn Model, Theorem 2, Section 4, and more clearly noting the reliance on Algorithm 1 in Guo et al. 2024. The authors should also fix the various minor issues (typos, math notation) mentioned by various reviewers.

The authors should also provide readers with better pointers to a fairly substantial amount of prior work on structural causal models in non-iid settings (e.g., Sherman 2022, Zhang et al. 2023, Jensen et al. 2020, Ogburn et al. 2014, Maier 2014). Perhaps the authors are only citing work that explicitly references exchangeability, in which case they should concentrate on that segment of the work that does that (e.g., Jensen et al. 2020, Ogburn et al. 2014). Note that these are only examples, and that the authors should do a more substantial literature review to find relevant prior work.

References

Jensen, D., Burroni, J., & Rattigan, M. (2020). Object conditioning for causal inference. Uncertainty in Artificial Intelligence (pp. 1072-1082). PMLR.

Maier, M. E. (2014). Causal Discovery For Relational Domains: Representation, Reasoning, And Learning. Doctoral dissertation, University of Massachusetts Amherst.

Ogburn, Elizabeth L, Tyler J VanderWeele, et al. (2014). Causal diagrams for interference. Statistical Science 29.4, pp. 559–578.

Sherman, E. (2022). Observational Causal Inference For Network Data Settings. Doctoral dissertation, Johns Hopkins University.

Zhang, C., Mohan, K., & Pearl, J. (2023, August). Causal Inference under Interference and Model Uncertainty.  Conference on Causal Learning and Reasoning (pp. 371-385). PMLR.